# Inference of long-range cell-cell force transmission from ECM remodeling fluctuations

Assaf Nahum[1], Yoni Koren[2], Bar Ergaz[2], Sari Natan[2], Gad Miller[1], Yuval Tamir[1], Shahar Goren[3], Avraham Kolel[2], Sankar Jagadeeshan [4], Moshe Elkabets [4], Ayelet Lesman [2,5✉] & Assaf Zaritsky [1✉]

Cells sense, manipulate and respond to their mechanical microenvironment in a plethora of physiological processes, yet the understanding of how cells transmit, receive and interpret environmental cues to communicate with distant cells is severely limited due to lack of tools to quantitatively infer the complex tangle of dynamic cell-cell interactions in complicated environments. We present a computational method to systematically infer and quantify long-range cell-cell force transmission through the extracellular matrix (cell-ECM-cell communication) by correlating ECM remodeling fluctuations in between communicating cells and demonstrating that these fluctuations contain sufficient information to define unique signatures that robustly distinguish between different pairs of communicating cells. We demonstrate our method with finite element simulations and live 3D imaging of fibroblasts and cancer cells embedded in fibrin gels. While previous studies relied on the formation of a visible fibrous 'band' extending between cells to inform on mechanical communication, our method detected mechanical propagation even in cases where visible bands never formed. We revealed that while contractility is required, band formation is not necessary, for cell-ECM-cell communication, and that mechanical signals propagate from one cell to another even upon massive reduction in their contractility. Our method sets the stage to measure the fundamental aspects of intercellular long-range mechanical communication in physiological contexts and may provide a new functional readout for high content 3D image-based screening. The ability to infer cell-ECM-cell communication using standard confocal microscopy holds the promise for wide use and democratizing the method.

[1] Department of Software and Information Systems Engineering, Ben-Gurion University of the Negev, Beer-Sheva 84105, Israel. [2] School of Mechanical Engineering, Faculty of Engineering, Tel-Aviv University, Tel-Aviv 69978, Israel. [3] Department of Biomedical Engineering, Faculty of Engineering, Tel-Aviv University, Tel-Aviv 69978, Israel. [4] The Shraga Segal Dept. of Microbiology, Immunology and Genetics, Faculty of Health Sciences, Ben-Gurion University of the Negev, Beer-Sheva 84105, Israel. [5] Present address: Center for Physics and Chemistry of Living Systems, Tel Aviv University, Tel Aviv 69978, Israel. ✉email: ayeletlesman@tauex.tau.ac.il; assafza@bgu.ac.il

Many types of cells apply considerable traction forces on their surrounding matrix leading to ECM remodeling that can propagate to large distances of tens of cell diameters away[1–15]. When embedded in fibrous biological hydrogels, such as collagen or fibrin, cells contract and thereby remodel and densify nearby ECM fibers. Then, in the time-scale of a few hours, these remodeling can form a visible fibrous band of aligned and dense fibers coupling neighboring cells mechanically which can influence the cells' internal molecular state[16] and active response[17]. This form of long-range cell-cell force

transmission through the ECM can be viewed as the imparting or exchanging of information between cells, and thus is aligned with the definition of communication[18] termed here cell-ECM-cell communication. This mode of long-range mechanical cell-ECM-cell communication was shown to coordinate various biological processes, including tissue injury[17], fibrosis[19], vascular assembly, capillary sprouting[1,14,20], tissue folding[21], and cancer invasion and metastasis[5,9]. In-vivo, fiber alignment bands can serve as ECM 'tracks' for cell migration with potential roles in wound healing, cancer metastasis and fibrosis[22,23].

Measuring the transfer of forces through the ECM during cell-ECM-cell communication is challenging and has been typically achieved indirectly by measuring changes in the density, alignment or displacement of the remodeled fibrous ECM between the cell pairs resulting from the active contraction of cells[4,5,9–11,13,24–26]. Most current measurements characterize the structure of the fibrous band extending between mechanically coupled cells to inform on cell–cell mechanical coupling, while relying on the visibility of a band between cells, formed when cell-generated forces are strong enough[5,9,11,13,26]. Such measurement lacks the sensitivity to measure the dynamic reciprocal mechanical information transfer between the cells by excluding potential cell-ECM-cell communication in the absence of visible bands, thus hampering our ability to distinguish which cells are actually communicating from the many cells that have the potential to communicate. Bridging this gap will enable tackling long-standing open questions in how tissues develop and diseases progress by enabling us to identify which cells are communicating with each other, and to what extent, in complex environments.

Here, we present a new computational method to quantify the transmitted ECM signal in between neighboring cells by correlating temporal fluctuations of the remodeled matrix. Computational simulations and 3D live imaging of fibroblasts and cancer cells embedded in fibrin gels demonstrate the power of our method in identifying unique ECM remodeling signatures that allow to robustly distinguish between different pairs of communicating cells. Using this method, we were able to measure communication between cell pairs that do not form a visible 'band' of densified ECM, and after partial depletion of contractility upon Myosin-II inhibition. These results imply that mechanical signals propagate from one cell to another even in low cell contractility levels.

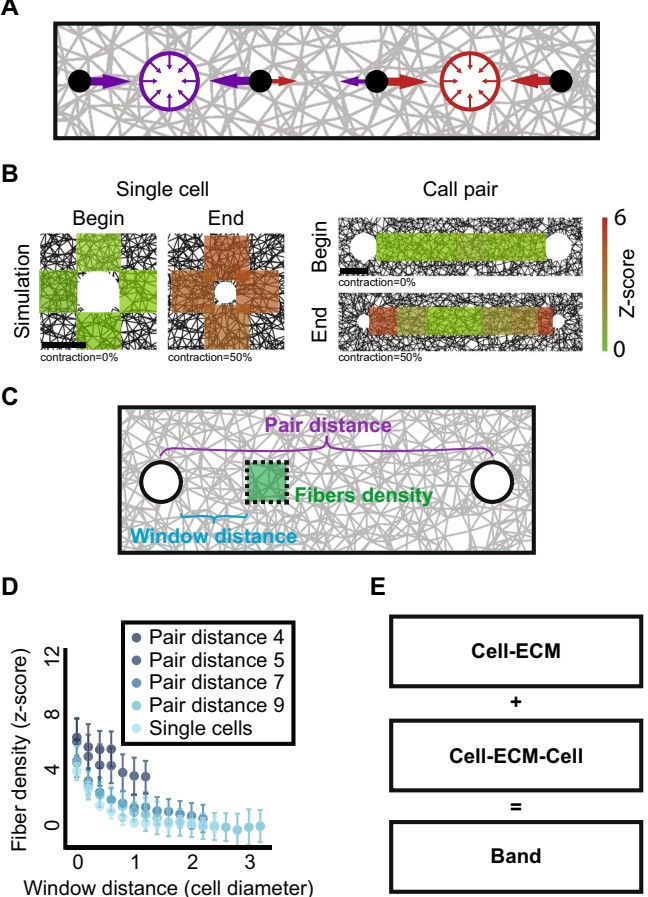

**Fig. 1 Quantifying ECM densification in simulated cell-ECM-cell mechanical communication. A** Fibers in between pairs of cells (red and purple circles) are remodeled by the integrated mechanical forces that both cells exert on the ECM. Colored arrows depict the magnitude of the force experienced in a specific location in the fibrous gel that are generated by the two cells. **B** Quantitative visualization of representative simulated cell-ECM interactions (left) and cell-ECM-cell communication (right) at the onset ("begin") and after ("end") 50% cell contraction. Color code encodes the ECM density in z-score, the number of standard deviations above background levels. Scale bar is one cell diameter. **C** Schematic sketch. Pair distance (purple) between cell pairs at the onset of imaging. Window distance (cyan) is measured from the boundaries of one cell toward the other cell in the pair, along the axis (in 3D) defined by the pair. **D** Quantifying fiber densification after 50% cell contraction as a function of the distance around simulated single cells ("single cells") and as a function of the distance between cell pairs ("pair distance"). Single cells ($N = 7$). Pair distance of 4 ($N = 19$), 5 ($N = 20$ pairs), 7 ($N = 20$ pairs), or 9 ($N = 19$ pairs) cell diameters. Note that window distance is bounded in lower pair distances. Mean and standard deviation are displayed for each distance. **E** A dense fibrous "band" between cell pairs is formed by the combined effects of cell-ECM interactions and cell-ECM-cell communication.

## Results

**Analysis of ECM density between pairs of communicating cells using finite element simulations.** During cell-cell mechanical communication, every localized fibrous region in between the neighboring cells is affected by two components: the contractile force of the adjacent cell (referred as 'cell-ECM' interaction), and the force transmitted from the second distant cell (referred as 'cell-ECM-cell' communication; Fig. 1A). To quantitatively characterize the independent contribution of each of these two components, we simulated contracting cells embedded within 2D fibrous networks using finite element discrete modeling, quantified the change in the ECM density between neighboring pairs and compared it to single cells interacting with the ECM. While these simulations do not reflect the true complexity of biological systems, they capture the essence of the mechanical elements of cell contractility and force propagation in fibrous nonlinear elastic networks[6,8,12,27–30] (Methods). Thus, these simulations serve as a minimal mechanical model that enables us to independently tune mechanical parameters (e.g., cell-cell distance and cell contractility), and quantitatively infer their effect on ECM remodeling. These simulations will be used to ask whether ECM

remodeling can robustly encode unique ECM signature in communicating cell pairs, forming the basis for our method.

To enable quantitative comparison between simulations, we normalized the fiber density to its z-score - the number of standard deviations away from the mean background fiber density at regions that were not influenced by the cells, implying positive z-scores values where the fibers are denser than average and negative values for regions that are less dense than the average background fiber density ("Methods" section). In simulations of single contractile cells, we found that the regions next to the cell's boundary became denser following the application of 50% cell contraction (Fig. 1B). Upon the presence of a second cell, the overall densification was governed by the integrated contractile activity of both cells (Fig. 1A). This dual contribution led to the formation of a band of increased density along the connecting axis between the cells (Fig. 1B–D). While for single cells the fiber densification faded to the background level at a characteristic distance of approximately 2 cell diameters, the bands between pairs of cells were characterized with increased fiber density that extended further away from the cells and were increased for cell pairs that were located closer to one another (smaller pair distance, Fig. 1D). The increased fiber density between cell pairs in comparison to single cells could potentially be used to quantitatively decouple the contribution of the interaction of the cell with the ECM (cell-ECM) and the mechanical information transmitted from the communicating partner (cell-ECM-cell) that together form the visible band between the cells (Fig. 1E). Specifically, we hypothesized that the information encoded by the cell-ECM-cell component that is unique to a specific communicating cell pair can be used to measure the long-range mechanical communication between cells.

**Distinguishing between communicating versus non-communicating simulated cell pairs using temporal correlation of local ECM remodeling fluctuations**. Given that local ECM regions located along a band experience forces exerted by both cells and the discrepancy between ECM remodeling by a single and a pair of cells (Fig. 1), we hypothesized that local ECM remodeling fluctuations contain sufficient information to distinguish communicating cells from cells that are not communicating with one another. We clarify that by denoting "communicating" cells we refer to two cells that are within pulling distance in the same (simulated) ECM network, while "non-communicating" cell pairs are two cells that come from different networks (Fig. 2A). To test this hypothesis, we simulated the temporal process that led to the final remodeling. We consecutively applied 1% cell contraction for 50 steps, reaching 50% cell contraction. For each step, we recorded the fiber density between communicating cells. These iterative simulations captured the temporal dynamics of force propagation between the cells during cell-ECM-cell communication. As expected, the fiber density close to the cells' edge gradually increased over time (Fig. 2B, C, Video S1).

Cell-ECM pulling occurs in noisy bursts of activity leading to fluctuations in ECM density that propagate to large distances. We hypothesized that these ECM remodeling fluctuations, measured in quantification windows adjacent to each cell, would temporally correlate between communicating cell pairs due to synchronized response of the mechanically coupled matrix, exceeding the temporal correlation between non-communicating cell pairs (Fig. 2A). In simulations of pairs of both communicating and non-communicating cells, we found positive correlations throughout the simulation in the fiber density dynamics and in its temporal derivative, i.e., change in fiber density over time (Fig. S1A). The correlation in fiber density is attributed to the

monotonic increase in ECM densification between communicating cells (Fig. 2C). The correlation in the temporal derivative of fiber density is attributed to an association between fiber density and the change in fiber density (Fig. S1B): the ECM became denser over time, leading to increased temporal derivatives. However, these general trends lead to high correlations between any two cells regardless of whether they communicate with each other, and confound the unique fluctuations that may be attributed to a specific pair of communicating cells. To overcome this confounder, we removed the temporal trends (detrending) via a second temporal derivative. This analysis pipeline transformed the raw simulated local ECM remodeling fluctuations to a correlation-based measure for cell-ECM-cell communication (Fig. 2D, full details in "Methods" section).

Another confounder was that every simulated cell contracted in the exact same magnitude as any other simulated cell leading to nearly exact time-dependent remodeling of the adjacent ECM regime and thus masking the unique ECM fluctuation patterns that may exist between communicating partners. Thus, our hypothesis was that variability in the cells' contraction activity (an inherent property of cells) will lead to variability in the local ECM remodeling around each cell, leading to a unique communication signature between partners. In other words, we thought that heterogeneity in ECM remodeling would enable us to decouple the general ECM remodeling that occurs between any random pair of cells from the specific signal that is transmitted between communicating partners. Indeed, when introducing heterogeneity in simulated cell contraction, we distinguished communicating from non-communicating simulated cell pairs while maintaining the non-communicating correlations around zero (Fig. 2E). Cell heterogeneity was included in the simulations by drawing the instantaneous contraction rate of each cell independently from a normal distribution with mean ($\mu$) contraction of 1% and a standard deviation ($\sigma$) in the range of 0-0.75% (std.; "Methods" section). Even a minimal standard deviation of 0.25% in cell contraction was sufficient to make a clear distinction between communicating and non-communicating cell pairs, which improved with increasing standard deviations (Fig. 2E). This distinction was negatively correlated to the pair distance - as cells were placed further apart, it became more difficult to distinguish between communicating and non-communicating cell pairs (Fig. 2F). Moreover, this distinction was improved when moving the ECM quantification window along the band, toward the communication partner (increasing the *window distance*), which increased the communication partner's influence (Fig. S1C). Finally, we verified that the correlation signal stems from the contractile activity of the two neighboring cells by demonstrating that the correlation of two contracting cell pairs exceeded the correlation of pairs of one contractile and a second non-contractile passive cell (Fig. S1D).

Altogether, these results established the notion that a correlation-based approach can capture force transmission between actively contractile cells, that temporal correlation of local ECM remodeling fluctuations can distinguish between communicating and non-communicating simulated cell pairs, and that contractile heterogeneity is required to make this distinction.

**Analysis of 3D ECM density between pairs of communicating fibroblast cells**. To test the simulated results of correlation in ECM remodeling as a marker of mechanical cell-ECM-cell communication, we used 3D time-lapse confocal microscopy to image live NIH/3T3 fibroblast cells (GFP-Actin) embedded in 3D fluorescently labeled fibrin gels, where fibrin intensity was used as a proxy of fiber density (Methods). Similar to the simulations, we

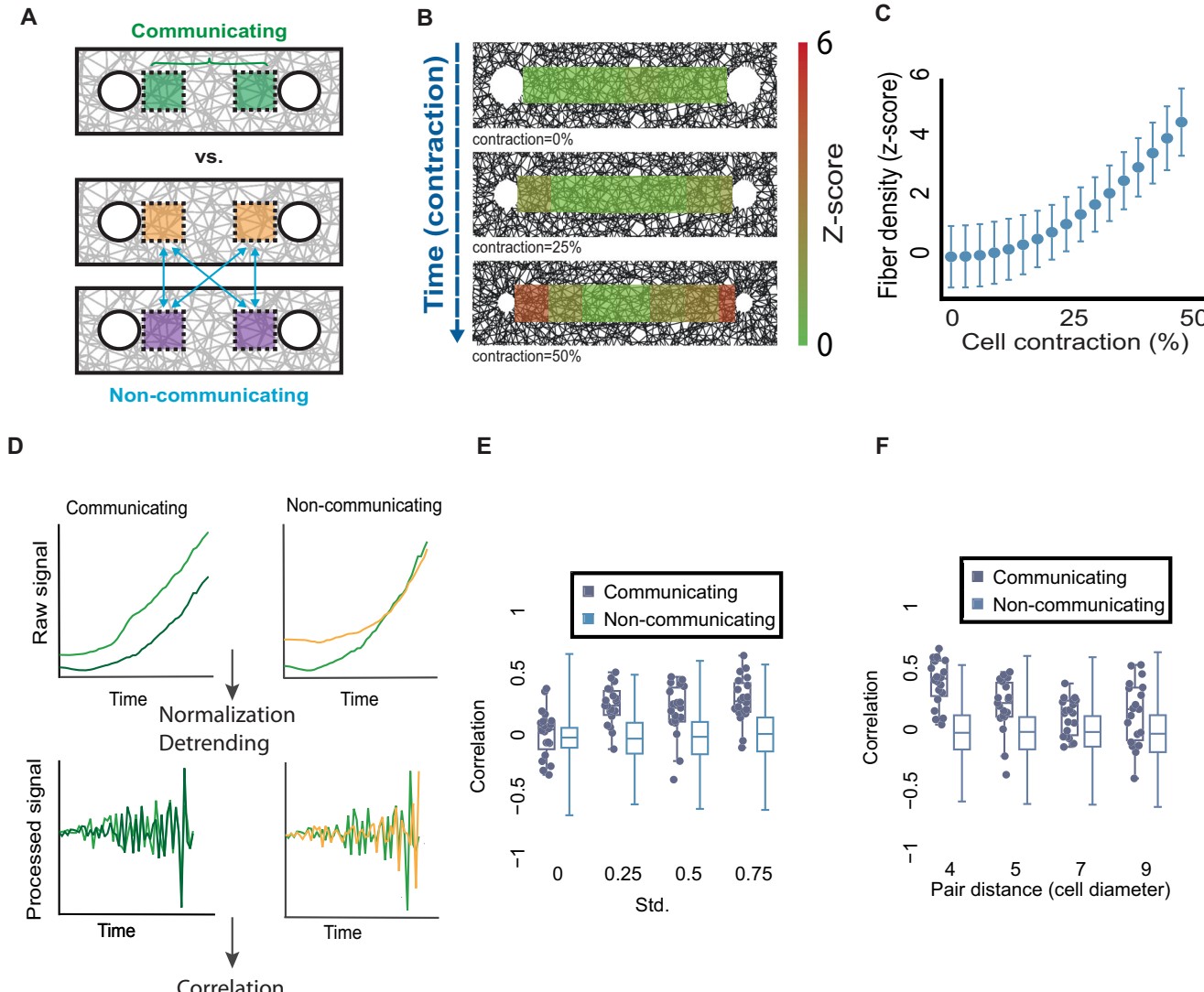

**Fig. 2 Using temporal correlations of ECM remodeling fluctuations to distinguish between communicating and non-communicating simulated cell pairs. A** Schematic sketch of comparing ECM remodeling fluctuations of communicating versus non-communicating cell pairs. The correlation between quantification windows of communicating pairs (upper in green) is evaluated in relation to the correlation between two cells from two different communicating pairs (lower in orange and purple). **B, C** Quantitative visualization and quantification of the dynamics of simulated cell-ECM-cell communication. **B** Representative simulation visualization at the onset (top), 25% cell contraction (middle), and 50% cell contraction (bottom). **C** Quantifying fiber densification as a function of simulated contraction steps in between cell pairs. $N = 20$ cell pairs. Color code (**B**) and y-axis (**C**) encode the ECM density in z-score, the number of standard deviations above background levels. The variability in fiber density is a consequence of the randomness in the network architecture. The pair distance between simulated cell centers was set to 7 cell diameters. **D** Correlation-based pipeline to measure cell-ECM-cell communication in simulations. Raw time series refers to measurement of ECM density in the quantification window (top), normalization with respect to the mean background fiber density at regions that were not influenced by the cells and detrending by second temporal derivative lead to a fluctuating signal (bottom) that is correlated to distinguish between pairs of communicating versus non-communicating cells (left versus right). **E** Distinguishing between communicating and non-communicating simulated cell pairs correlations of the fluctuations of the second derivative of fiber density with different levels of contraction heterogeneity. Each data point records the correlation between a simulated communicating cell pair, each box plot record the corresponding distribution of correlations between (the many) pairs of non-communicating cells. Mean contraction of 1% and standard deviation (heterogeneity) of 0%, 0.25%, 0.5%, and 0.75%. Three statistical tests are performed according to the following order. (1) Wilcoxon sum-rank testing the null hypothesis that the two correlation distributions of communicating and non-communicating cell pairs are originating from the same distribution. Wilcoxon sign-rank testing the null hypothesis that the correlation distributions of (2) communicating or (3) non-communicating are distributed around a mean 0. Std. = 0%: $N = 20$, p-values: (1) not significant, (2) not significant, and (3) not significant. Std. = 0.25%: $N = 19$, p-values: (1) < 0.0001, (2) < 0.001, and (3) not significant. Std. = 0.5%: $N = 19$, p-values: (1) <0.0001, (2) <0.001, (3) not significant. Std. = 0.75%: $N = 20$, p-values: (1) <0.0001, (2) <0.001, (3) not significant. **F** Correlation of communicating versus non-communicating cell pairs using the second derivative of fiber density dynamics as a function of the pair distance. Pair distance: 4 ($N = 20$ pairs, p-value < 0.0001), 5 ($N = 19$ pairs, p-value < 0.0001), 7 ($N = 19$ pairs, p-value < 0.01), and 9 ($N = 19$ pairs, p-value < 0.01). Data points for non-communicating cell pairs were not displayed in panels **E**, **F** because there are too many of them, while the number of data points for communicating cells is much smaller and thus can be plotted.

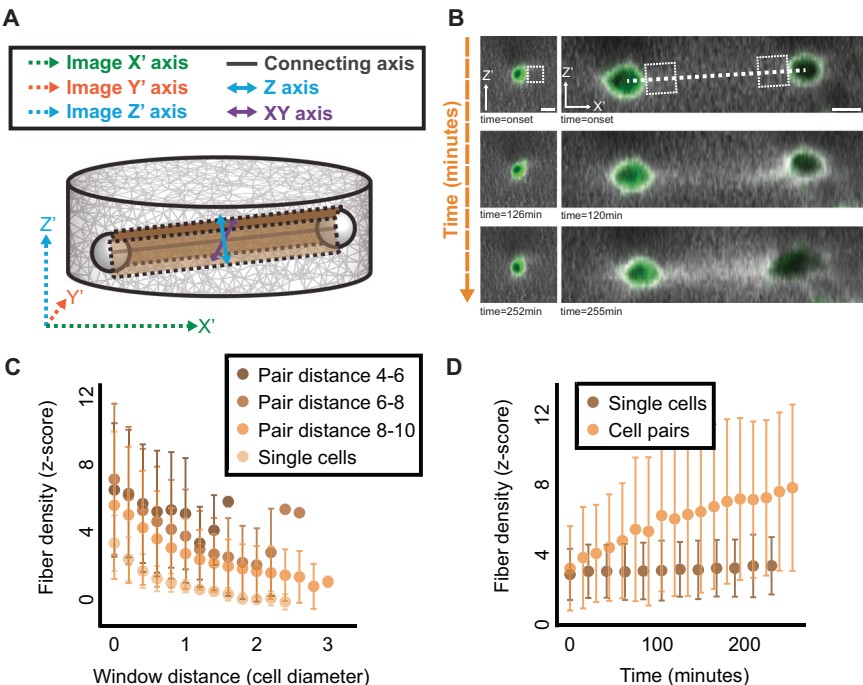

**Fig. 3 Quantifying ECM densification during cell-ECM-cell mechanical communication with 3D live imaging of communicating fibroblast cells. A** ECM remodeling dynamics are quantified along the connecting axis between the cells: illustration of the top-side view. Microscopy axes (X′, Y′, Z′) marked with dashed arrows, transformed visualization and quantification axes (connecting axis, XY, Z) marked with solid arrows. The cuboid region in-between the cell pair (brown), has the width of a cell diameter, and is used for quantification along the connecting axis. The length of the cuboid region is the pair distance. The cuboid left and right sides were parallel to the microscopy axial plane. The Z-axis (cyan) and the XY-axis (purple) are perpendicular to each other and to the connecting axis (black). Side view and top view are illustrated in Fig. S2. **B** Representative images of a single cell (left) and a cell pair (right) at the onset (top), after (slightly over) two hours (middle), and (slightly over) four hours of live cell imaging (bottom). The initial distance between the cell centers in the pair was ~117 μm (~7.8 cell diameters assuming mean fibroblast diameter of 15 μm). White dashed line represents the connecting axis between the cells. White dashed rectangle next to the cell boundaries represents one quantification window that is used in panels **C**, **D** (see "Methods" section for details). Scale bar = 15 μm. **C** Quantifying cell-ECM-cell communication after (slightly over) four hours of live cell imaging as a function of distance between cell pairs, using the window size shown in B. Single cells ($N = 7$), Pair distance of 4–6 ($N = 9$ pairs), 6–8 ($N = 13$ pairs) and 8-10 ($N = 5$ pairs) cell diameters (assuming cell diameter of 15 μm). **D** Quantifying fiber densification dynamics near a single cell and "band" formation in between pairs of cells, using the window size shown in B. $N = 7$ single cell, $N = 13$ cell pairs at pair distances of 6–8 cell diameters (90–120 μm). Error bars in **C**, **D** indicate standard deviation.

clarify that "communicating pairs" are pairs of cells located next to each other in the fibrous gel and thus are within a pulling distance, whereas "non-communicating cells" are pairs that are not in proximity (beyond 10 cell diameters away, see Methods). To visualize and measure fiber intensity in between cell pairs in 3D, we transformed the microscopy axes to a new 3D coordinate system that is aligned around the connecting axis between the cells' centers (Figs. 3A and S2 and Video S2, "Methods" section) and performed z-score fiber intensity normalization in respect to background quantification windows defined at the onset of the experiment at regions that were not influenced by the cells ("Methods" section). While single cells did not show visually apparent fiber densification beyond regions close to the cell, approximately 60% of all cell pairs formed a visible band of increased density extending along the connecting axis between the cells (Fig. 3B) that showed increased mechanical coupling for cells pairs that were closer to one another (Figs. 3C and S3). The fiber density close to the cells' edge gradually increased over time in pairs of communicating cells but remained constant in single cells (Fig. 3D, Video S3 versus Video S4). The higher fiber intensity at the onset of imaging was attributed to the time (approximately 30 min) that passed from setting up the experiment until the onset of imaging (Fig. 3D, z-score of ~3 standard deviations above the background intensity). During this time, cells have already remodeled the fibers around them. These

results conclude that the formation of dense fibrous bands between fibroblast cell pairs embedded in 3D fibrin gels are indicative of a mechanical coupling in concurrence with previous studies[5,9–11].

**Temporal correlation of local ECM remodeling fluctuations defines a signature for cell-ECM-cell communication.** Rather than comparing communicating versus non-communicating cell pairs (Fig. 2E), definitive quantification of cell-ECM-cell communication lies in the ability to distinguish between pairs of communicating cells, i.e., whether the ECM fluctuations of one cell pair has a unique signature that can be distinguished from that of a different pair of communicating cells. We tackled this challenge by testing whether the correlation between a simulated pair of communicating cells surpassed the correlation between one cell from that pair and another cell from a different simulated pair of communicating cells located in a different fibrous network (Fig. 4A). Specifically, we compared the correlation in the second temporal derivative of the local ECM density dynamics between quantification windows adjacent to each cell pair, located in the 'same' pair versus a cell in a 'different' pair, and accordingly coined the term *same-versus-different pair analysis* (schematic in Fig. 4A). Having the "same" correlation exceeding the "different" correlation implies that the correlation between a communicating cell pair is not merely an effect of a similar ECM densification

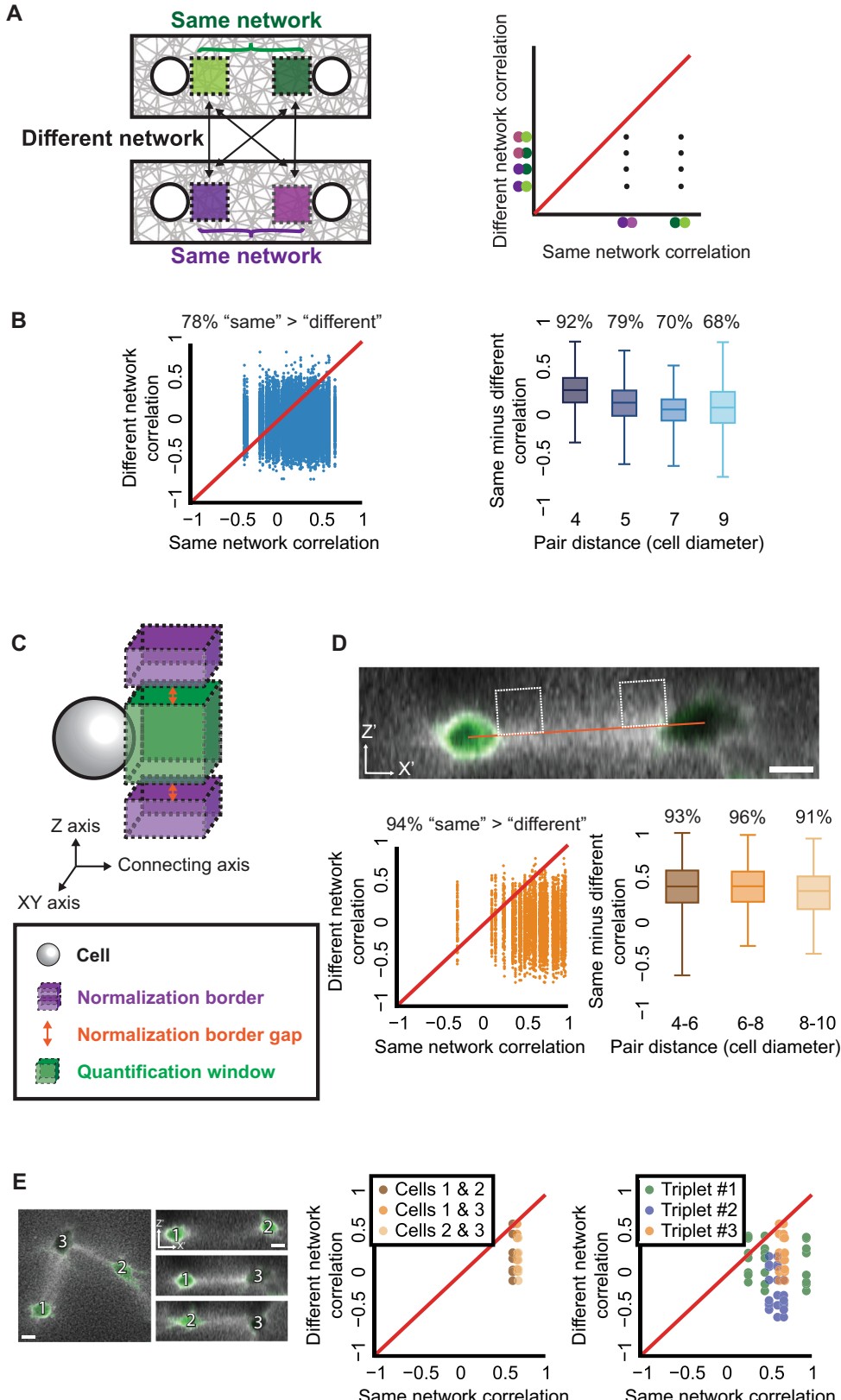

pattern that is common to any pair of communication cells, but rather is indicative of communication unique for the "same" cell pair at test. We considered all possible ordered combinations of triplets of cells that include a pair of communicating cells and a third cell that takes part in a different communicating cell pair (Fig. 4A). We then analyzed these triplets using the "same"

(communicating) versus the matched "different" (non-communicating) cell pair correlations. In the corresponding plot (Fig. 4B), each data point represents the correlation of one communicating cell pair (*x*-axis) to multiple non-communicating cells (*y*-axis), each leading to a different value (shown as a vertical line of points in Fig. 4B). Data points below the diagonal y = x

**Fig. 4 Distinction between pairs of communicating cells with same-versus-different pair analysis. A** Schematic sketch of the same-versus-different pair analysis. Left: the correlation between quantification windows of communicating pairs ("same", green or purple) is evaluated in relation to the correlation between one cell from that pair and another cell from a different communicating pair ("different", purple and green). Right: example of *same-versus-different pair* results plot, with only two pairs of cells. *X*-axis is the "same" pair correlations, *Y*-axis is "different" pair correlation. The red line is $X = Y$, thus every data point below that line implies that the "same" correlations are stronger. **B** Same-versus-different pair analysis in simulations. Each data point records the correlation between the "same" and "different" cell pairs using the second derivative of fiber density dynamics. Mean contraction of 1% and standard deviation of 0.5%. All combinations of "same"/"different" were considered. Left: Pair distances: 4 ($N = 20$), 5 ($N = 19$), 7 ($N = 19$), and 9 ($N = 19$) cell diameters. "Same" pair had a higher correlation than "different" pair in 78% of the matched correlations. Right: "Same" - "different" correlation distributions for different pair distances: 4 ($N = 20$ pairs), 5 ($N = 19$ pairs), 7 ($N = 19$ pairs), and 9 ($N = 19$ pairs) cell diameters. 92% (pair distance = 4), 79% (pair distance = 5), 70% (pair distance = 7) and 68% (pair distance = 9) of data points were positive, implying that the "same" correlation is higher than the "different" correlation. Wilcoxon signed-rank testing the null hypothesis that the "Same" - "different" correlations were distributed around a mean 0: $p$-value < 0.0001 for all pair distances. "Same" – "different" correlations were different for cell pairs at distances of 4 versus 9 cell diameters (T-test $p$-value < 0.001). **C** Schematic sketch of a cell, the quantification window and the normalization border. The normalization border is composed of two parts, above and below the quantification window, each with a width of approximately a cell diameter (15 μm), height of 0.5 cell size and with a gap of 0.25 from the quantification window (full information in "Methods" section). **D** Same-versus-different pair analysis in experiments. Top: representative pair of communication cells shown after 255 min of imaging. Scale bar = 15 μm. Line connects the cell centers in the XY/Z space. Quantification windows were placed ~0.5 cell diameter (7.5 μm) above the connecting axis between cell pairs with a visible band, no offset in X/Y-axis. The same image is also shown in Fig. 3B. Middle: analysis pipeline: raw time series were normalized in respect to local regions close to their quantification window to reduce local spurious correlations (see panel **C**), and correlations were calculated using the first derivative of fiber density dynamics, which was sufficient to remove non-stationarity effects in ECM remodeling fluctuations. Full details in Methods. For the displayed cell pair (top) the Pearson correlation coefficient was 0.66 (Z offset = 0) and 0.71 (Z offset = 0.5). Bottom left: pair distance = 60-150 μm (~4-10 cell diameters). $N = 48$ cell pairs, all combinations of "same"/"different" were considered. "Same" pair had a higher correlation than "different" pair in 94% of the matched correlations. Bottom right: "same" - "different" correlation distributions for different pair distances: 4–6 ($N = 18$ pairs), 6–8 ($N = 19$ pairs), and 8–10 ($N = 11$ pairs) cell diameters. "Same" pair had a higher correlation than "different" pair in 93% (pair distance 4–6), 96% (pair distance 6–8), and 91% (pair distance 8–10). Wilcoxon signed-rank test $p$-value < 0.0001 for all pair distances. **E** Same-versus-different pair analysis for cell triplets. Quantification windows were placed ~0.5 cell diameter (7.5 μm) above the connecting axis between the cells. Correlations were calculated using the first derivative of fiber density dynamics. Left: triplet of cells communicating with one another after 255 mins of live cell imaging. Scale bars = 15 μm. Middle: same-versus-different pair analysis for the cell triplets from the left panel. Pair distances approximately 7.3 (Cells 1 vs. 2), 6.3 (Cells 1 vs. 3), and 4.9 (Cells 2 vs. 3) cell diameters. "Same" pair had a higher correlation than the "different" pair in 7/8 (Cells 1 and 2), 8/8 (Cells 1 and 3), and 8/8 (Cells 2 and 3) of the matched correlations. Wilcoxon signed-rank test $p$-value < 0.05. Right: Multiple triplets. $N = 3$ triplets color coded in shades of green, blue and orange. Triplet #3 was presented in the previous panels (matched shades of orange between panels). 92% of matched same minus different correlations were positive. Wilcoxon signed-rank $p$-value < 0.0001.

(red line in Fig. 4B) indicate that correlation between the communicating pair exceeded that of the non-communicating pair. In simulated cells, a "same" pair had a higher correlation than a "different" pair in 92% of the matched correlations for pair distance of 4 cell-diameters, and this correlation gradually reduced with increased pair distance (Fig. 4B). To assess the validity of the method for fibrous ECM network with varying mechanical properties we also simulated a network with a relative linear-elastic behavior until 15% strain compared with the current fiber stiffening model (Fig. S4A). Same-versus-different analysis was improved for the stiffening model (Fig. S4B), implicating the role of strain stiffening in long-range mechanical communication, in line with our previous studies[12, 30,31]. Altogether, simulated cells communicating with one another were more synchronized in their ECM remodeling fluctuations, and same-versus-different analysis could distinguish between different pairs of communicating cells.

We next aimed at extending our computational analysis of simulated data to distinguishing between pairs of communicating cells in experimental data. Our analysis pipeline was adjusted to enable analysis of experimental data. The first temporal derivative was sufficient to remove non-stationarity effects in ECM remodeling fluctuations of single and pairs of communicating fibroblast cells (Fig. S5), so we could avoid the second derivative in the correlation analysis of experimental data. Unlike simulations, all cells in a single experiment were embedded in the same fibrous gel, possibly inducing spatial ECM correlations in nearby regions that do not necessarily involve cell-ECM-cell communication. Thus, we had to control for spurious correlations that could originate from the proximity between the ECM regions in the same network. This was achieved by including an additional step in the quantification where we corrected the ECM remodeling fluctuations in the quantification window in respect to local regions close to that window to reduce local temporal artifacts that may lead to erroneous correlations (Fig. 4C, "Methods" section).

We performed the same-versus-different pair analysis, but were able to quantitatively distinguish between different pairs of communicating fibroblast cells only after shifting the quantification window 7.5 μm above or below the axis connecting the communicating cells (Figs. 4D and S6). A "same" pair had a higher correlation than its corresponding "different" pair in 94% of the matched correlations, for various pair distances (Fig. 4D and S7A–C) and even for different cell pairs within triplets of cells (Fig. 4E).

To control for potential masking of cell-ECM-cell communication by correlated non-communication related local ECM remodeling we devised a computational control that considered ECM regions (without cells) located close to a communicating cell, which we term *real-versus-fake pair analysis*. We created new pairs where each is composed of one real cell and another fake cell (cell-free ECM region) located in the exact same distance as the matched pair of communicating cells, and away from other cells (Fig. 5A, "Methods" section). ECM remodeling correlation between a pair of communicating cells ("real-real") was higher than its corresponding "real-fake" pair in 86% of the matched correlations providing a standardized internal control and further validation that our method truly captures cell-ECM-cell communication (Fig. 5B).

Given our ability to distinguish one pair from a different pair of communicating cells, we wondered whether we can match a cell to its true communication partner when considering all other cells in the experiment (Fig. 5C, Video S5, Methods). For this "matchmaking" analysis, we considered the true communication

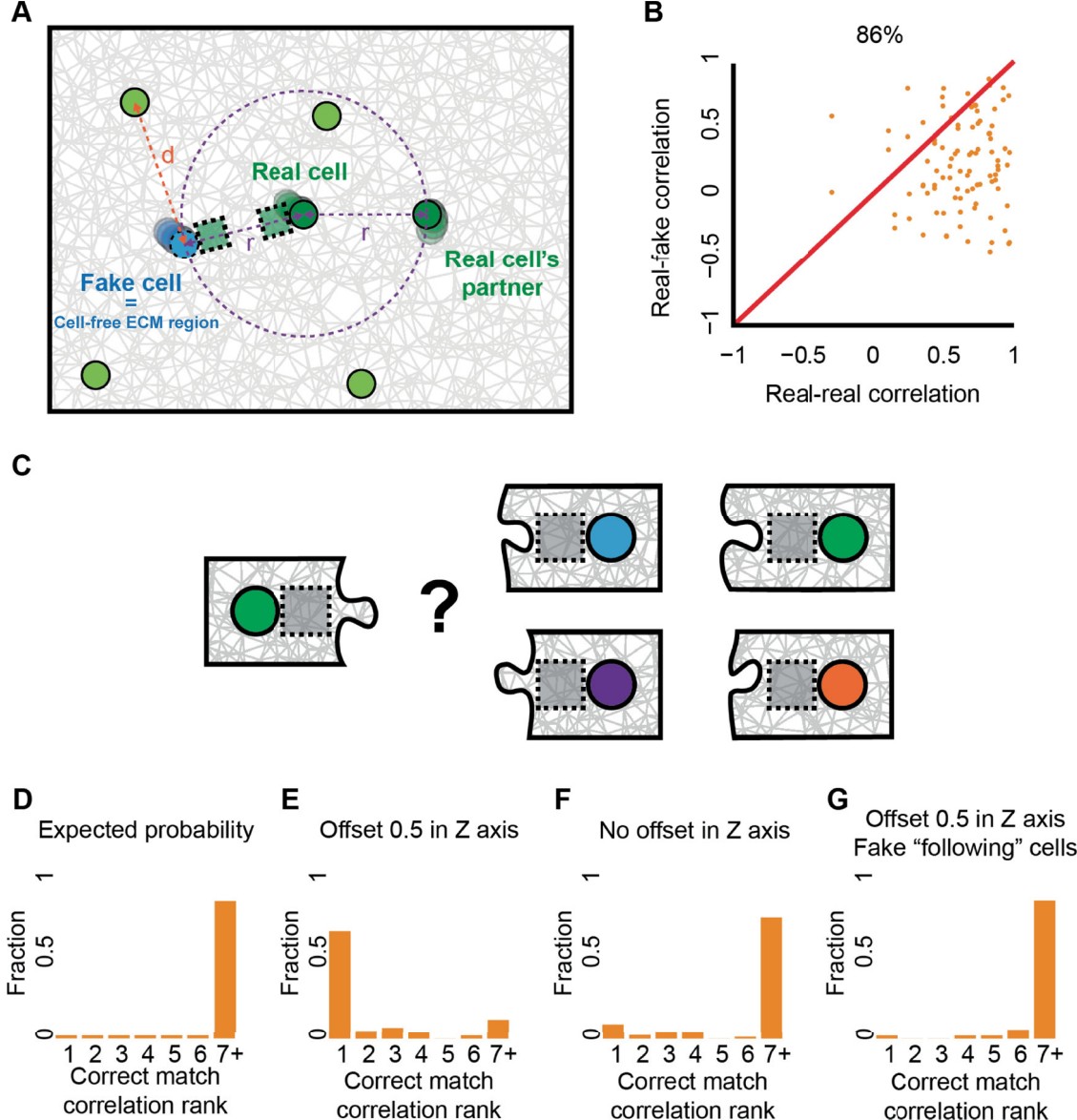

**Fig. 5 Quantitative identification of unique ECM remodeling signatures of communication partners by quantifying the confounding factor of non-communication-related local remodeling in the ECM and matchmaking between communication partners. A** Schematic sketch of the real-versus-fake pair analysis. "Fake" cell (cyan) coordinates were located on the circumference of a circle (dashed purple), with a radius *r* from the corresponding "real" communicating cell (dark green), and with the maximal distance (*d*, orange) in respect to all other real cells (light green). **B** Correlations between ECM fluctuations of pairs of communicating cell pairs ("real-real", x-axis) versus the correlation between a cell and non-cellular-related local remodeling in the ECM ("real-fake", y-axis). Quantification windows were placed ~0.5 cell diameter (7.5 μm) above the connecting axis between the communicating cells, no offset in X/Y-axis. Correlations were calculated using the first derivative of fiber density dynamics. $N = 48$ "real-real" cells at cell pair distances ranging 4–10 cell diameters. "Real-real" pair had a higher correlation than "real-fake" pair in 86% of cases. Wilcoxon sign-rank *p*-value < 0.0001. **C** Schematic sketch of the matchmaking analysis. ECM remodeling dynamics of a given cell (left) was correlated with 49 other cells (with repeats – see "Methods" section) from an experiment (right). The rank of the correlation with the true communication partner (i.e., the position in the sorted list of all correlations with other cells, marked in green) was recorded. This process repeated for all cells. **D–G** Distributions of the correlation rank with the true communication partner for cell pairs with a visible band. Pair distance of 60–150 μm (~4–10 cell diameters). Correlations were calculated using the first derivative of the ECM intensity over time. "Correct match" refers to a correlation rank of 1. **D** Random matching: simulation of arbitrary matching leads to correct matching probability of 0.02 ($N = 50$ cells). **E** Quantification window 0.5 cell diameters (7.5 μm) above the connecting axis between the cells: correct matching probability of 0.69 (selecting from $N = 96$ cells), compared to 0.02 for random matching (panel **D**). **F** Quantification window at the connecting axis between the cells: correct matching probability of 0.1 (selecting from $N = 84$ cells), compared to 0.02 for random matching. **G** ECM regions (without cells) located close to each other ("fake-following" pairs, see Fig. S8 for schematics): correct matching probability of 0.02 ($N = 50$ regions), compared to 0.02 for random matching.

partner as the one that is connected with a visible band. For each cell, in every cell pair with a visible band, we recorded the first derivative of its fiber density dynamics, correlated it to all other cells in the gel, and reported the rank of its true matching partner. With 50 potential partners, the expected random probability of identifying the communication partner is $1/50 = 0.02$ (Fig. 5D). Using correlation-based matching, the probability of identifying the true communication partner was 0.69 (Fig. 5E). This accuracy of identifying the matching partner dropped to 0.1 when considering quantification regions along the connecting axis between the cells (i.e., no offset in the z-axis, Fig. 5F) and was not attributed to correlated non-communication related local ECM remodeling as verified by careful analysis that considered ECM regions (without cells) located close to each other ("fake" pairs, Figs. 5G and S8, "Methods" section).

To support the generalization of our method in measuring cell-ECM-cell communication in cell systems beyond fibroblasts we performed experimental validation with a different cell model of murine cancer Hras-transformed mutated cells[32] (Fig. S9, "Methods" section). Faster imaging (5 min per frame) provided a more sensitive readout for cell-ECM-cell communication, reaching perfect accuracy in same-versus-different, real-versus-fake, and matchmaking analyses, thus enhancing the capacity to distinguish pairs of communicating cells (Fig. S10). However, in considering the inherent tradeoff between temporal resolution and the number of cell pairs we can image in a single experiment with our microscope we prioritize the latter for purposes of collecting sufficient statistics for the rest of this study.

Altogether, we defined three computational measurements providing us with systematic means to decipher cell-ECM-cell communication: (1) Same-versus-different – quantifying the ability to distinguish between ECM remolding of one pair of communicating cells from a different pair of communicating cells, (2) Real-versus-fake – quantifying the confounding factor of non-communication-related local remodeling in the ECM, and (3) Matchmaking – quantitative identification of unique ECM remodeling signatures of communication partners. True communication is characterized by high values in all three measurements.

**Experimental controls validate ECM remodeling correlations as a measurement for cell-ECM-cell communication.** To verify that ECM-remodeling correlations are robust to ECM fluctuations induced by other cells in the gel we included a new set of experiments as negative controls. Severe reduction in all three communication readouts was measured between pairs of fluorescent beads in size comparable to cells, and in experiments with dead cells (Fig. 6A, B), compared to live fibroblasts (Figs. 4 and 5) and cancer cells (Fig. S9). To include background ECM remodeling resulting from contraction of other live cells, we co-cultured live and dead cells within the same gel. In the presence of live cells, no communication was measured between pairs of dead cells (Fig. 6C) and between live-dead cell pairs (Fig. 6D) - a validation that our measurement for cell-ECM-cell communication is robust to local ECM remodeling originating from contractile activity of the two partners and not from other cells in the gel or remote sensing of physical boundaries through the ECM (e.g.[33]) correspondingly. A summary of all measurements over all experimental conditions are reported in Table S1.

**Decoupling band formation and communication sensing.** Our finding that cell-ECM-cell communication can be identified quantitatively only when shifting the quantification window away from the connecting axis between the cell centers, encouraged us to systematically measure the relations between cell-ECM-cell

communication and band formation in experiments. We analyzed different ECM quantification windows in the 3D space around the cell by shifting them laterally (perpendicular, in the XY axis) and axially (up and down, in the Z-axis) in relation to the connecting axis (Fig. 7A, "Methods" section). For each shifted window we measured the final normalized fiber density after 255 mins and also, independently, performed same-versus-different pair analysis for the corresponding windows over time. While the fiber density was maximal along the connecting axis, the discrimination between different pairs of communicating cells was optimized above or below (axially) the connecting axis (Fig. 7B–D), and excluded the possibility that this was an artifact of saturated pixels along the dense fibrous band (Fig. S11). These results were consistent for faster imaging experiments (Fig. S12) establishing that communication sensing, i.e., our method's ability to identify cell-ECM-cell communication, is decoupled from band formation.

Previous studies analyzed the intercellular band as a measure of mechanical communication. The decoupling between band formation and communication sensing led us to examine whether our method can measure communication for cell pairs that did not form a visible band between the cells (Fig. 7E–G). Indeed, same-versus-different, real-versus-fake and matchmaking analyses identified cell-ECM-cell communication for band-less pairs, slightly above or below the connecting axis, implying that communication is present even when a band is not visible to the naked-eye (Figs. 7H and S13, and Video S6). Faster imaging identified more prominently cell-ECM-cell communication for cell pairs without a visible band (Fig. 7I).

We conclude that our method's ability to measure communication is improved when correlating quantification windows located slightly away from the connecting axis in-between the communicating cells, while the densest fiber band is formed directly along the connecting axis and that cell-ECM-cell communication exists even when no visible band is formed between the cell pair. These results challenge the current notion that band formation is the hallmark of cell-cell communication in fibrous gels.

**Higher signal-to-noise ratio off the connecting axis supports sensitive measurement of cell-ECM-cell communication.** We hypothesized that cell-ECM-cell communication is best measured slightly away (in the Z-axis) from the connecting axis between the cells because the higher ECM intensities along the fibrous band confound our ability to sensitively measure the ECM fluctuations as a signature of cell-ECM-cell communication. To test this hypothesis, we calculated the association between fiber density and the sensitivity in measuring communication via same-versus-different pair analysis by correlating these parameters across different locations of the quantification window. This analysis demonstrated a sweet spot for measuring cell-ECM-cell communication (Fig. S14A). Regions with low fiber density, that were located far away from the connecting axis, had low same-versus-different discrimination, and this low discrimination was also apparent in denser band regions along the connecting axis. Analysis of the temporal change in fiber density revealed that while the band intensity is maximal along the connecting axis between the cells, the magnitude of the changes in ECM intensity is similar along the connecting axis and slightly above it (Fig. S14B). Because the communicating signal is measured as the change in ECM intensity relative to the background intensity (Fig. 4C), the 'signal to noise ratio' of the measurable communication signal is higher off the connecting axis, making it a more sensitive measurement for cell-ECM-cell communication.

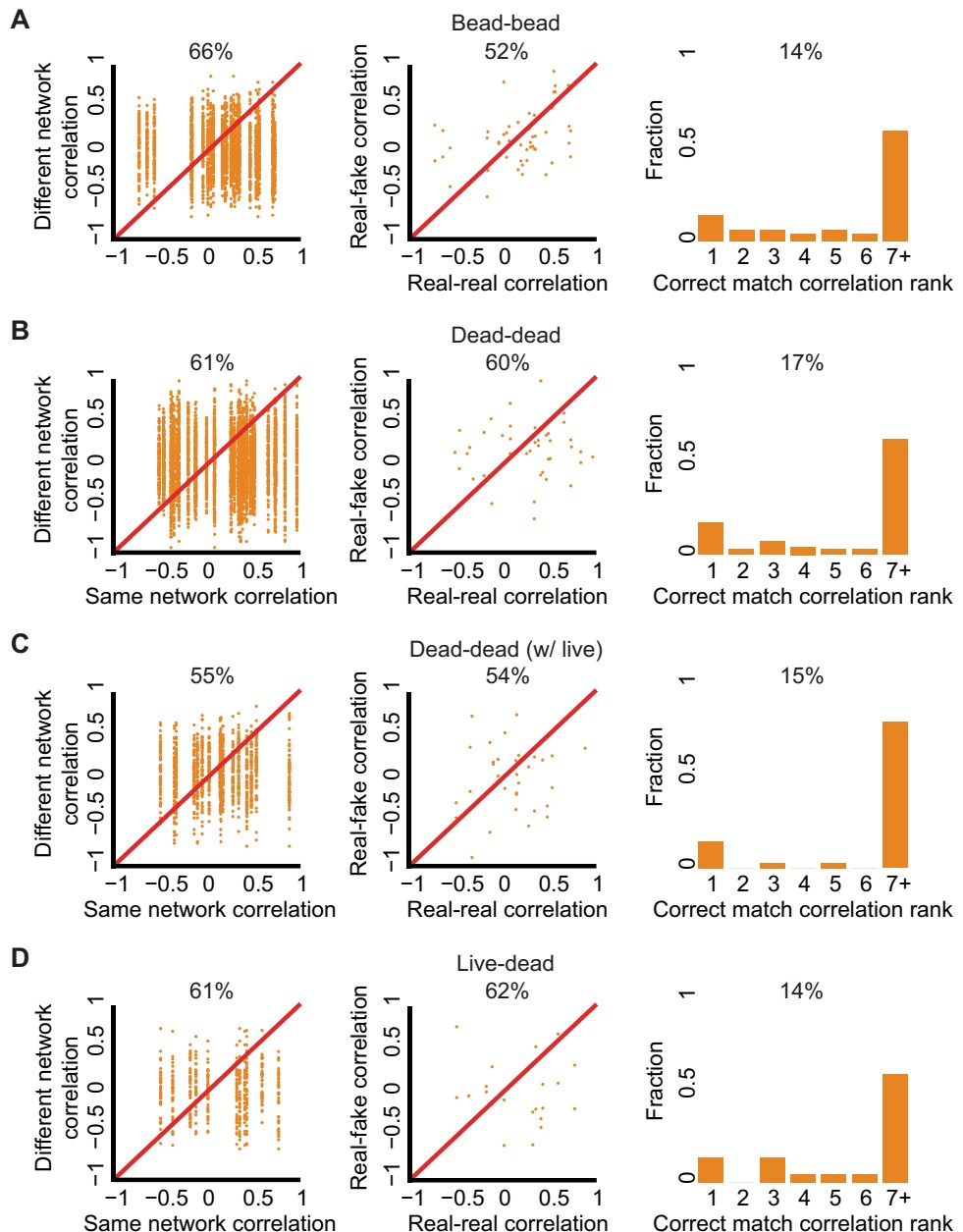

**Fig. 6 Experimental negative controls measured with same-versus-different analysis (first column), real-versus-fake analysis (middle column), and match making analysis (right column). All three measurements were significantly reduced in all the negative control experiments. A–D** compared to live fibroblasts (Figs. 4 and 5) and cancer cells (Fig. S9). **A** Experiments with fluorescent beads. Same-versus-different: $N = 25$, 66% "same" > "different, p-value < 0.0001. Real-versus-fake: $N = 25$, 52% "real" > "fake", p-value not significant. Matchmaking analysis: $N = 50$, correct matching probability = 14%. **B–D** Experimental controls with live and dead fibroblast cells. **B** Dead cells. Same-versus-different: $N = 35$, 61% "same" > "different, p-value < 0.0001. Real-versus-fake: $N = 29$, 60% "real" > "fake", p-value not significant. Matchmaking analysis: $N = 70$, correct matching probability = 17%. **C** Dead cells with the presence of live cells. Same-versus-different: $N = 17$, 55% "same" > "different, p-value < 0.0001. Real-versus-fake: $N = 17$, 54% "real" > "fake", p-value not significant. Matchmaking analysis: $N = 34$, correct matching probability = 15%. **D** Live and dead cells. Same-versus-different: $N = 17$, 61% "same" > "different, p-value < 0.0001. Real-versus-fake: $N = 11$, 62% "real" > "fake", p-value not significant. Matchmaking analysis: $N = 22$, correct matching probability = 14%.

**The role of Myosin II contractility on cell-ECM-cell communication.** Given that we could measure cell-ECM-cell communication even without a visible band forming between the cells, and considering the fact that this type of long-range communication is inherently mechanical, we next asked what is the role of contractility in cell-ECM-cell communication. Contractility inhibition with a dosage of 85 nM of Blebbistatin, a Myosin II inhibitor, revealed that cell-ECM-cell communication can be measured even when contractility is partially inhibited and a

visible band between the cells practically never forms (Fig. S15A). Further increasing the Blebbistatin dosage to 150 nM verified that contractility is required for this mode of mechanical communication (Fig. S15B). Cumulatively, these results suggest that mechanical signals propagate from one cell to another even upon massive reduction in their contractility and thus cells can mechanically communicate even with reduced contractility levels that are not sufficient to form a visible band.

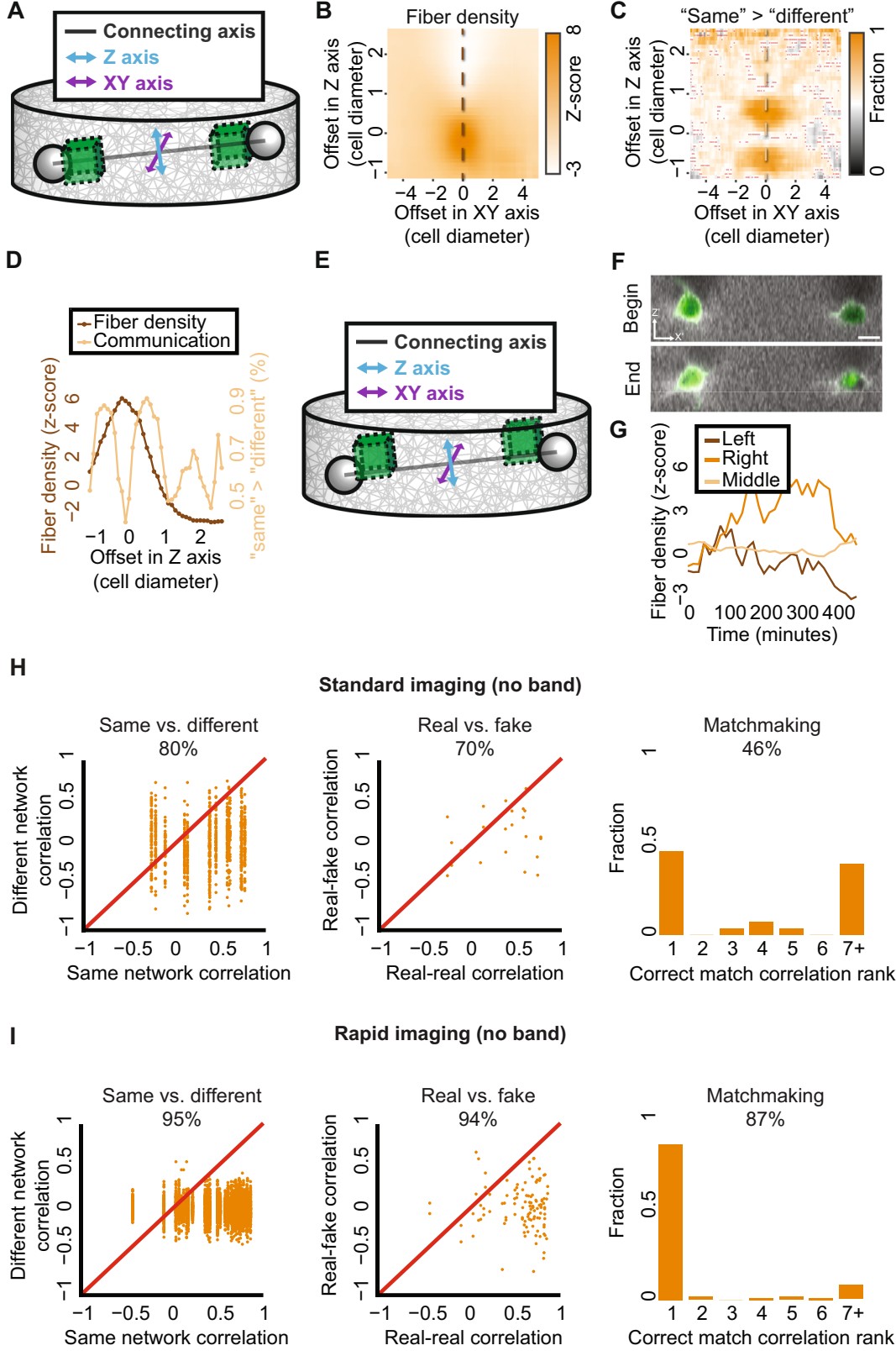

## Identifying leader and follower in pairs of simulated communicating cells.

We next asked if we could use our approach to quantify asymmetric interactions between the communicating partners. More specifically, can we identify which cell in a communicating pair is more dominant or influential? Our quantitative interpretation of "influential" is that past ECM-remodeling fluctuations of one cell are predictive of the future ECM-remodeling of its communicating partner. Such temporal order defines a leader-follower relation. We performed simulations where a "follower" cell "imitates" the previous contraction of its influential "leader cell". In other words, the contraction of both cells are determined by the leader cell with a time lag of one

**Fig. 7 Spatial decoupling of band formation and cell-ECM-cell communication. A** Cell pair axes schematic sketch. Z-axis (cyan) and XY-axis (purple) are perpendicular to each other and to the connecting axis between the cells' centers (black). **B–D** Mean fiber density and cell-ECM-cell communication (same-versus-different analysis) for shifted quantification windows in cell pairs with a visible band. Pair distances of 60–150 μm (~4–10, cell diameters). The dashed vertical line in panels B-C is used as a line profile in panel **D**. **B** Fiber density. Mean fiber density for systematic offsets in Z and XY axes. **C** Cell-ECM-cell communication. Mean fraction of higher "same", correlation between communicating pairs, versus "different", correlation between one cell from that pair and another cell from a different communicating pair for systematic offsets in Z and XY axes. Red 'x' marked that the null hypothesis that "same" - "different" correlations are distributed around zero was not rejected with p-value ≤ 0.05. **D** Fiber density and cell-ECM-cell communication along the axial line-scan (offset in XY axis = 0). Peaks in cell-ECM-cell communication appear above (and below) the connecting axis between the cells, where fiber density is maximal on the connecting axis. **E** Schematic sketch of the optimal quantification window location to quantify cell-ECM-cell communication. Offset in Z axis = 0.5. Offset in XY axis = 0. **F–I** Cell-ECM-cell communication for cell pairs with no visible band. Quantification windows were placed 7.5 μm (~0.5 cell diameter) above the connecting axis between the cells. **F** Representative cell pair with no visible band at the onset ("begin") and after ("end") 255 mins of cell imaging. Scale bar = 15 μm. **G** Quantification of the dynamics of the first derivative of ECM intensity in the cell pair in panel **F** implicating the absence of a band. Left/right - quantification windows adjacent to cell boundaries, "middle" - quantification window in between the cells centers at each time frame. Pearson correlation coefficient between left and right cells inner regions = 0.7, p-value < 0.0001. **H**, **I** Cell pairs with no visible band with same-versus-different analysis (first column), real-versus-fake analysis (middle column), and match making analysis (right column). **H** Standard time resolution of 15 mins per frame. Same-versus-different: N = 14, 80% "same" > "different, p-value < 0.0001. Real-versus-fake: N = 15, 70% "real" > "fake", p-value < 0.01. Matchmaking analysis: N = 28, correct matching probability = 46%. **I** Rapid time resolution of 5 mins per frame. Same-versus-different: N = 56, 95% "same" > "different, p-value < 0.0001. Real-versus-fake: N = 56, 94% "real" > "fake", p-value < 0.0001. Matchmaking analysis: N = 112, correct matching probability = 87%.

simulation step (Fig. S16A). The simplest approach to quantitatively identify a temporal order is by cross-correlation with time lags, where the correlation between two time-series is calculated for a given lag, and the time-lag that leads to the maximal correlation determines the temporal order. This analysis successfully identified which simulated cell was the leader and which one was the follower with an accurate lag time (Figs. S16B and S17A). To evaluate the robustness of using correlations to identify leader-follower relations we simulated cell pairs where the follower contraction was composed of an independent component, and a component that is dependent on its leader: $Contraction_{follower}(t) = (1-\alpha)*N(\mu, \sigma) + \alpha*Contraction_{leader}(t-1)$, where t > 1, $0 \le \alpha \le 1$, $\mu = 1$, $\sigma = 0.5$ are the mean and standard deviation correspondingly, and $Contraction_{leader}$ is drawn from the normal distribution $N(\mu, \sigma)$. The term $\alpha$ indicates the "followership" magnitude, higher values of $\alpha$ imply that the follower is more influenced by its leader contraction. The correlation increased with increased influence of the leader cells (Fig. S16C) and the maximal cross correlation occurred at the correct lag, accurately identifying the leader/follower roles for all simulated pairs for $\alpha = 0.5$, where the follower cell contraction is determined with equal contribution from its intrinsic "decision" and the extrinsic influence by its leader (Figs. S16D and S17B). In a second validation we tested whether we can identify leader/follower when the follower contracts more than its leader. This leads to increased ECM remodeling that might propagate to the leader cell and mask its influence on the follower. We set $\alpha$ to 1 (follower is copycatting the leader's contraction), and introduced $\beta \ge 1$ – the fold increase of the follower's contraction: $Contraction_{follower}(t) = \beta * Contraction_{leader}(t-1)$. The correlation was nearly identical for $\beta \le 1.2$ (20% increase in follower contraction), and the first mistaken prediction of the follower/follower assignment occurred for $\beta = 1.2$, for 1 out of 7 simulated pairs (Figs. S16E and S17C). We further validated these results by pairing leaders or followers to cells from other simulated communicating pairs to create artificial pairs of cells that did not interact with one another (Fig. S18). However, cross-correlation analysis did not identify leader-follower relations in experimental data (Fig. S19). This inability to identify leader-follower relations in experimental data could be attributed to lack of sufficient temporal resolution – the response of the follower cell may occur in time scales faster than the 5 min temporal resolution in this study. Another alternative is that our method is not sufficiently sensitive to measure the subtle ECM fluctuations that distinguish

leader from follower cells. There is always the possibility that fibroblast cells do not form leader-follower communication patterns, perhaps due to insufficient heterogeneity that can be resolved by assessment of cells pairs from mixed genetic backgrounds. These possibilities are left to be explored in future studies. Cumulatively, these results verified that cross-correlation of ECM remodeling fluctuations can robustly identify leader and follower cells in simulated communicating cells but not in our experimental data.

## Discussion

Our work proposes a systematic computational method to quantify cell-ECM-cell communication and demonstrates its applicability in simulations and experiments. The method quantifies the local fiber remodeling dynamics between pairs of communicating cells, located up to a distance of 10 cell-diameters apart, in 2D (simulations) and 3D (experiments) by applying the following key steps. Normalization in relation to the background to enable robust comparison across experiments; Subtracting background remodeling to avoid masking of the communication signal; Detrending to avoid spurious correlations; Systematic evaluation of the location of the quantification window to extract the most informative signal; Measurement of the temporal correlation as a quantitative readout for cell-ECM-cell communication. Our method provides technical advances that will open the door for cell biologists and biophysicists to decipher how cells process mechanical information transmitted through the microenvironment.

We combined finite element simulations and 3D live cell imaging experiments. Our minimal model, although not reflecting the true complexity of the biological system, captures the essence of cell contraction and force propagation in fibrous nonlinear elastic networks[6,8,12,27,29,30] and thus serves as a viable tool to control various parameters independently to test and verify the sensitivity and robustness of our approach. For example, examining the minimal cell contractile heterogeneity required to effectively quantify cell-ECM-cell communication (Fig. 2E) and measure its decay as a function of the distance between the cells (Fig. 4B), assessing the effect of changing ECM material from linear to nonlinear behavior (Fig. S4) and testing the method's sensitivity to cell autonomously and contraction magnitude that may mask the capability to identify leader-follower relations (Fig. S16B–E).

We found that heterogeneity in cell contractility created unique temporal patterns of ECM fluctuations that were necessary to quantitatively identify cell-ECM-cell communication, allowing us to distinguish between different pairs of communicating cells using temporal correlation of local ECM remodeling fluctuations. We devised two measurements for cell-ECM-cell communication (same-versus-different and matchmaking) and a third measurement to decouple non-communication-related local ECM remodeling from real communication (real-versus-fake). These in silico controls combined with experimental validations verified that our method can sensitively and robustly measure cell-ECM-cell communication in fibroblast and cancer cells located up to a distance of ~8–10 cell-dimeters apart (Figs. 4, 5, and S9).

Until now, the consensus in the field was that the existence of a visible fibrous band is indicative of cell-ECM-cell communication in fibrous gels, like collagen or fibrin[5,9,11,13]. Our method does not rely on the existence of a band, rather it correlates local ECM remodeling as a measure for "communication", the unique mechanical signal propagating from one cell to its communication partner. We demonstrate the sensitivity of correlating ECM fluctuations by establishing that the formation of a denser ECM region between the cells is not required for cell-ECM-cell communication, thus decoupling fiber densification and long-range mechanical intercellular communication. In fact, we find that the fibrous band is a confounder to measure the unique communication signature between a pair of cells (Fig. 7), probably due to lower 'signal to noise ratio' (Fig. S14). Moreover, dose-dependent Blebbistatin experiments revealed that while cellular contractile force is required for communication, cells are still able to communicate even after substantial reduction of their contractility, where fibrous bands never form (Fig. S15). These results align with previous studies suggesting that myosin II–mediated contractility acts as an inhibitor for cell-to-cell mechanical communication by arresting passive mechanical force transduction through the cellular cortex during collective cell migration[34–37].

We aimed at identifying leader-follower relationships within pairs of communicating cells. We simulated a situation where one cell influenced its communicating partner, determining its future contractions and demonstrated that our method can robustly identify the leader and follower cells from the simulated ECM-remodeling fluctuations. This was even possible in challenging scenarios where followers were only partially influenced by their leader or contracted up to 20% more than their leader (Fig. S16C–E). These results set theoretical limitations on our method's capacity to decouple cell contractility from leadership status due to increased ECM remodeling that can propagate to the leader cell and mask its influence on the follower. We were not able to identify leader-follower relations in our experimental data, either because of lack of sensitivity in the given experimental conditions (e.g., time resolution, subtle "leadership" signal in cell pairs from the same genetic background) or because this mode of matrix-mediated mechanical communication does not yield leader-follower pairs in fibroblasts. However, our computational model of follower-leaders can motivate future experiments to investigate the notion of leader-follower relations between cells in long-range mechanical cell-cell communication, for example under conditions as faster imaging and/or co-culturing cells in different molecular/functional states or from different genetic backgrounds.

The ECM remodeling fluctuations are relatively small. Noise and measurement errors, for example due to low spatial or temporal resolution, might deteriorate the communication signal and hamper our ability to measure cell-ECM-cell communication. While this is a possibility, validations in our experimental settings, by artificially reducing temporal (Fig. S10) or spatial (Fig. S21) resolution demonstrated that the communication signal

is quite robust. Moreover, we used standard confocal imaging in this study, which is inherently limited in its temporal resolution, axial resolution and size of field of view. Despite these limitations, we succeeded to quantitatively characterize intercellular mechanical communication through fibrous environments, being able to near perfectly match all communication partners, with tens of potential candidates for each match, only from ECM remodeling fluctuations. This performance, especially given the standard microscopy that is available in almost any academic institute, highlights the robustness of our approach and its potential of democratizing cell-ECM-cell communication quantification.

Our proof of principle study is an enabler of mechanistic understanding of long-range cell-cell mechanical communication, and sets the ground for potential applications. Having the ability to quantitatively measure cell-ECM-cell communication, could enable to systematically probe the cellular and molecular players for this mode of mechanical communication, naturally starting with the actomyosin cytoskeleton and adhesion complexes[16]. We expect that our method could be generalized to non-fibrous environments, such as synthetic (e.g., poly-acrylamide hydrogels) or biological (e.g., Matrigel) hydrogels, by embedding tracer particles that will enable the indirect measurement of local ECM remodeling fluctuations. Another exciting venue is deciphering the role of long-range intercellular communication in the more complex and physiological relevant microenvironments in vivo. A third extension of our method would be to apply it to other modes of cell-cell communication. For example, using intracellular molecular fluctuations as the functional readout to infer cell-cell communication, an approach previously taken in[38] to reveal signaling ordering at the intracellular scale.

One application where the ability to precisely measure cell-cell communication may be useful is tissue engineering. By controlling the patterning of multiple cell types one could optimize tissue formation according to cell type specific communication capabilities[39]. Another application is high content 3D image-based screening. Image-based phenotypic screening is traditionally applied with 2D imaging, quantifying single cell morphology and distributions of intracellular fluorescent intensities, and is applied for multiple applications including identification and characterization of small molecules in drug discovery[40]. Recent tools enable high-content 3D image-based cell phenotyping[41], providing a more physiologically relevant context for in vivo follow-up studies[42]. The interplay between 3D cell morphology, the interactions with the environment and the mechanical communication with other cells will likely provide important complementary functional readouts for 3D image-based phenotypic cell phenotyping that are not accessible with current methods. Such phenotyping could be very useful for applications where the cellular microenvironment and the interactions between cells and the ECM are established hallmarks, such as in cancer or fibrosis[22,23,43–45].

## Methods

### Computational modeling

*Finite element simulations of cell contraction in fibrous networks.* We used our previously described computational finite element model of one or two cells contracting in two-dimensional fibrous networks[12,30,31]. We used the finite element software Abaqus/CAE 2017 (Dassault Systèmes Simulia) to model the network mechanics and perform the simulations. The software's standard\implicit solver was used in all simulations. The cells were modeled as circular voids in the fiber networks. The fibers were represented by discrete one-dimensional elements connected by nodes (Fig. S20A), randomly distributed to set an isotropic and homogeneous network structure prior to cell contraction (see next). The output of each finite-element simulation included the information regarding each network element's location and dynamics.

*Fiber network architecture.* We used Matlab R2018b to construct the network geometry and architecture as previously described[12]. The networks were designed to optimize the fiber orientations distribution toward uniformity (i.e., isotropic) and toward homogeneous fiber density. Briefly, we devised a random process to create network geometries, as we previously reported in ref. [12]. The process starts from uniformly scattering nodes in a circular domain. The nodes were then connected by fiber elements by considering an objective cost function which controls the fiber length, fiber connectivity (i.e., the mean number of fibers intersected at each node) and the angle between fibers connected at each node. The network coordinates spanned from −2 to 2 (AU) in X- and Y-axis (Fig. S20B). Cell centers were located along the X-axis, with a cell diameter of 0.08 (Fig. S20B). The mean fiber thickness was 0.2 AU, and mean fiber length was 20 AU, fitting typical fiber density-to-length ratio for collagen\fibrin gels[29,46,47]. The cell diameter before contraction was set to 0.08 AU, so that the cell diameter/mean fiber length ratio was 4:1, a typical ratio for fibroblast cells embedded in fibrin gel[47]. The average connectivity of the network was set to eight, to balance the tradeoff between the finite element software numerical stability and physiological relevance. We note, however, that the mean connectivity of collagen networks is typically lower and within the range of 3–4[48]. In our previous numerical studies, we found that the main characteristics of force transmission by contractile cells are maintained in the low and higher connectives networks[12,30].

*The mechanical properties of the simulated fiber networks.* Fibers were connected to one another by nodes, which acted as a freely rotating hinge, allowing for a rotation of the fibers without resistance. The fibers were modeled as linear truss elements, undergoing uniaxial tension or compression, without bending. They were characterized by nonlinear behavior typical to ECM fibers (such as collagen), including buckling under compression[6,8] and stiffening under tension[4]. We represented the buckling of the fibers by an elastic modulus which was ten times smaller at compressive strains exceeding 2% relative to the elastic modulus at small strains (−2 to 2%). Stiffening was achieved by an exponential increase in the elastic modulus for tensile strains larger than 2%[12,47,49]. In all simulations, the outer boundary of the network was fixed for translations and rotations.

To assess the validity of the method for fibrous ECM networks with varying mechanical properties, we performed additional simulations in which we tested the effect of changing the stiffening behavior to account for a material with a relative linear behavior until 15% strain, representative of fibrin gels (Fig. S4A), based on, for example ref. [50].

*Simulating cell-ECM-cell communication.* Cell contraction was modeled by applying a boundary condition of radial isotropic contractile displacements to all nodes constituting the cell boundaries, reaching up to 50% contraction of the cell radius. To simulate time we consecutively applied 1% cell contraction for 50 steps, reaching a final 50% cell contraction. Heterogeneity in cell contraction was implemented by applying cell contraction selected from a normal distribution with a mean ($\mu$) of 1% and varying standard deviations ($\sigma$) of 0, 0.25, 0.5, or 0.75, in each simulation step. In all simulations, the network size was set to 50 cell diameters and the cells were placed in its center to prevent boundary effects.

*The role of the simulations as a minimal mechanical model for extensive assessment of our method.* The simulations capture the essence of the material mechanical properties, mechanical elements of cell contractility and force propagation in fibrous nonlinear elastic networks. However, the representation of cells as circular cavities that contract uniformly (with unrealistic high contraction) in 2D without responding to the propagated mechanical signal, is far from realistic biology. Still, this minimal implementation captures the essence of the experimental system in terms of its mechanical aspects (contractile units in fibrous nonlinear networks) and thus serves as a "clean" mechanical framework to examine a process that is inherently mechanical (propagation of forces between cells). Thus, these simulations allow us to obtain proof-of-concept to the ability of a correlation-based method to detect a mechanical signal that propagates in the ECM between two contractile elements. Within this framework, the simulations unambiguously demonstrate the power of correlation in detecting the propagation of a mechanical signal from one cell to the other, as well as the effect of cell-cell distance, cells contraction mismatch and the ability to detect leader-follows. Revisiting the minimal model to include components that are more realistic is not central to the current methodology study, and will be addressed in future studies.

*Leader-follower simulations.* These simulations were implemented such that one cell (the "follower") contracted with a dependency on the other cell (the "leader"). The leader contraction at each time point was selected from a normal distribution $N(\mu = 1\%, \sigma = 0.5\%)$. In the first time step, the follower contraction was drawn independently, from the same normal distribution. From the second step and onwards, the follower contraction was composed of two components:

$$Contraction_{follower}(t) = (1 - \alpha) * N(\mu, \sigma) + \alpha * Contraction_{leader}(t - 1)$$

where $Contraction_{follower}(t)$ is the follower intrinsic contraction at time step $t$ drawn from $N(\mu = 1\%, \sigma = 0.5\%)$ and $Contraction_{leader}(t - 1)$ is the leader's contraction at the previous time step drawn from the same distribution. $\alpha$ is a constant that

varies between 0 and 1 and defines the level of leader-follower dependency: $\alpha = 0$ is the case of independently contracting cells and $\alpha = 1$ is the case where the follower cell repeats the leader's contraction with 1 frame delay (see Fig. S17A). In the second leader-follower simulation, the follower contracts according to the following equation:

$$Contraction_{follower}(t) = \beta * Contraction_{leader}(t - 1)$$

where $\beta > 1$, thus imitating the leader with a certain augmentation.

## Experiments

*Cell culture and chemical reagents.* Swiss 3T3 fibroblasts stably expressed GFP-actin (obtained as gifts from S. Fraser, University of Southern California, Los Angeles, CA) were cultured in DMEM supplemented with 10% fetal bovine serum, non-essential amino acids, sodium pyruvate, L-glutamine, 100 units/ml penicillin, 100 µg/ml streptomycin, and 100 µg/ml Neomycin, in a 37 °C humid incubator.

Murine Hras mutated-transform cells have been generated by infecting normal epithelial cells of the tongue with mutated-HRAS and shTP53 as described in[51].

Fluorescent beads experiments: 10–14 µm polystyrene beads (SPHERO™, Spherotech, Inc., Lake Forest, IL, USA) were embedded in 20 µl fibrin gels (10 µl thrombin, 10 µl fibrinogen) in concentration of $8 \times 10^3$.

Dead cells experiments: dead cells were prepared by placing cells in a hot water bath at 65 °C for 30 min, $8 \times 10^3$ dead cells were mixed in 20 µl of fibrin (10 µl thrombin, 10 µl fibrinogen).

Co-culture live and dead cells within the same gel: $4 \times 10^3$ dead cells were mixed with $4 \times 10^3$ live cells. After gel polymerization, ethidium homodimer (5 µM, red fluorescence) was added to the medium and incubated for 30 mins at 20–25 °C. This resulted in staining the nuclei of dead cells.

Blebbistatin experiments: 2 ml medium containing 85 µM or 150 µM blebbistatin were added to cover the polymerized cellular gel.

*Fibrinogen labeling.* Alexa Fluor 546 carboxylic acid, succinimidyl ester (Invitrogen) was mixed with fibrinogen solution in a 7.5:1 molar ratio for 1 h at room temperature and then filtered through a HiTrap desalting column (GE Healthcare) packed with Sephadex G-25 resin, to separate the unreacted dye.

*3D fibrin gel preparation.* GFP-Actin 3T3 fibroblast cells ($8 \times 10^3$ cells) were mixed with 10 µl of a 20 U/ml thrombin solution (Omrix Biopharmaceuticals). Then, 10 µl of a 10 mg/ml fluorescently labeled fibrinogen (Omrix Biopharmaceuticals) suspension was placed in a 35-mm cover-slip bottom dish (MatTek Corporation) and mixed gently with the 10 µl cells suspended in thrombin. The resulting fibrin gel was placed in the incubator for 20 min to polymerize, after which, a warm medium was added to cover the gel. The fibrin gels had an approximate shape of half a sphere, attached to the bottom surface of a cover slip, with a gel height of ~ 2-3 mm, and cells were fully embedded in the 3D gel. From this point on, the gels including the cells embedded in them, were maintained in 37 °C 5% CO₂.

*Time-lapse confocal microscopy.* Pairs of cells were imaged with a Zeiss 880 confocal microscope, equipped with a 40X NA = 1.1 water immersion lens (Zeiss) and a 30 mW argon laser was used to image both the cells (GFP-Actin) and the fluorescently-labeled-fibrin matrix with excitation wavelength of 488, and a separated emission spectrum for each. Throughout imaging, the gels with the contained cells were maintained in a 37 °C 5% CO₂ incubation chamber. Confocal z-stacks were acquired every 5-15 min for about 6 hours from cell seeding. We manually validated that the imaged cells did not undergo division to avoid the enhanced contraction during division. Confocal imaging inherently includes a tradeoff between the temporal resolution, the axial resolution and the number of imaged locations: higher temporal/axial resolution leads to the lower numbers of locations imaged leading to smaller numbers of communicating cell pairs per experiment. Experiments were imaged in three settings: (1) Temporal resolution of 15 mins, with 21 locations of images in resolution of 512 × 512 pixels (0.41 × 0.41 × 2 µm in X′Y′Z′) and 36 Z-slices. (2) Temporal resolution of 5 mins, with 9 locations in each and resolution of 256 × 256 pixels (0.83 × 0.83 × 2 µm in X′Y′Z′) and 40 Z-slices. (3) Temporal resolution of 21 mins, with 7 locations of images in resolution of 512 × 512 pixels (0.41 × 0.41 × 0.53 µm in X′Y′Z′) and 187 Z-slices, where X′, Y′ and Z′ are the microscopy axes. The spatial image resolution used in our study is not sufficient to localize individual fibers, but fine resolution is not required for our method, which uses the time series of the fluorescent intensity accumulated in a quantification window. We prioritized imaging large field of views over better spatial or temporal resolution to increase the number of cell pairs we can image in a single experiment and thus enhancing our statistical power. Moreover, the use of relatively low temporal and spatial resolutions are in fact an advantage. The ability to robustly detect cell-ECM-cell communication even at lower resolutions is yet another indication for our method's sensitivity and the wide availability of standard confocal imaging in practically any academic institute further highlights the potential of democratizing cell-ECM-cell communication quantification.

## Image analysis and quantifications

*Preprocessing live imaging data.* We first applied Fiji's[52] "Bleach Correction" on the raw fiber channel with the "Histogram Matching" option. On the actin (cell)

channel we applied a median filter (radius = 2) followed by a Gaussian blurring filter (sigma = 2), before segmenting the cells over time using the "3D Objects Counter" Fiji's plugin[53] (Threshold = 15, Size filter > 400). The cell's center coordinates in 3D for each time frame was recorded and used for cell tracking. Custom Python code was used for the cell tracking, by identification of cells to track in the first time frame and simply assigning the nearest cell (in 3D Euclidean distance) in the next time frame to construct the trajectory. Shorter trajectories were recorded for cells that moved beyond the field of view during imaging. This simple approach was sufficient thanks to the sparsity of the cell seeding.

*Transforming 3D images for visualization and quantification.* To visualize and quantify the 3D band between a cell pair we transformed the image to a new coordinate system that is defined in relation to the spatial relation between the pair. We transformed the image from the microscopy axes (denoted X′,Y′,Z′) to the following three new axes. The connecting axis, defined by the line connecting the cells' centers (Fig. 3A, Fig. S2 and Video S2 black line). The Z axis, parallel to the microscopy axial plane (Z′) and perpendicular to the connecting axis (Fig. 3A, Fig. S2, and Video S2 cyan line). The XY axis, perpendicular to the connecting axis and to the Z axis (Fig. 3A, Fig. S2, and Video S2 purple line). For visualization, we used the new 2D axis defined by XY and Z (Fig. S20C). We used Fiji's[52] "Reslice" function (default "Output spacing", "Slice count" according to the fibroblast diameter of 15 μm) to slice the images from top to bottom in the XY axis perpendicular to the connecting axis between the cells, with a width of 15 μm, interpolating axial pixel values to match the spatial resolution in XY using bilinear interpolation. Finally, we averaged the pixel intensities across the slices using Fiji's "Z project" (Projection type = "Average intensity") to create the 2D visualization. To visualize single cells, we picked an arbitrary XY axis (either 0°, 45°, 90° or 135°) with the same axial axis (Z). These visualizations were used for all experiments and all manual annotations (identifying imaging artifacts, and cell pairs with/without a visible band).

We implemented custom Python code to quantify the ECM density between a pair of communicating cells. First, we transformed the 3D axes to XY slices and Z, replicating the visualization without the Z-interpolation and without the slices averaging. Second, we performed another transformation, rotating the image onto the connecting axis between the cells to reach a common Z-axis. This transformation generates a 3D image where the Z-axis is perpendicular to the XY-axis that is perpendicular to the connecting axis between the cell's centers. This property allows us to move axially in relation to the 3D line connecting the cells. The whole process is depicted in Fig. S20C.

The second transformation rotated the original X′, Y′ and Z′ axes thus defining a new coordinate system, where the transformed pixel size ("resolution") in the new connecting axis and in the Z axis are a weighted combination of the original microscopy resolution in X′, Y′ and Z′. $ConnectingAxis_{resolution} = \frac{\theta}{90} * Z'_{resolution} + \frac{90-\theta}{90} * X', Y'_{resolution}$; $Z_{resolution} = \frac{\theta}{90} * X', Y'_{resolution} + \frac{90-\theta}{90} * Z'_{resolution}, 0 \le \theta \le 90)$, where $\theta$ is the calculated rotation angle between the connecting axis and the microscopy lateral plane before the rotation. The XY axis resolution remains unchanged ($X', Y'_{resolution}$). These new resolutions were used to calculate the size and location of the quantification window (see below).

Single cells do not have a preferred axis in 3D since they do not have a communicating partner. Thus, to quantify ECM density near single cells we sampled around each cell in 32 different orientations in 3D. 16 transformations were defined using all paired combinations of four angles (0°, 45°, 90°, and 135°), each transformation pair was applied similarly to the transformations in cell pairs. For example, the pair <45,135> implies first rotating the image at 45° in X′, Y′ (blue arrow in Fig. S20D) followed by a 135° rotation in Z′ (green arrow in Fig. S20D). These 16 transformations were used in two directions along the rotated axes leading to 32 orientations for quantification (see below).

*Manual filtering of defective image frames.* A small fraction of frames in a few experiment locations had imaging-related artifacts that hampered our ability to accurately segment the cell and quantify ECM densities. These artifacts included incorrect cell segmentation, dark areas due to imaging malfunctioning of the microscope and "light waves" (Video S7) that may have been the result of an air or water bubble trapped in the lenese's immersion oil. To include these experiments in our analysis we manually identified and recorded defective frames that had these artifacts and considered only valid time frames (without this artifact), when computing fiber density and correlations.

*Manual annotation of cell pairs with or without a visible band of increased density.* For analysis we considered cell pairs with pair distance ranging at 60–150 μm (4–10 cell diameters, assuming fibroblast diameter of 15μm). Based on previous studies that focused solely on cell pairs with a visible band of increased density between them[6,9,11,49], we partitioned our dataset to cell pairs that formed and ones that did not form a visible band of denser fibers between the cells. This partition was performed manually by visual assessment of the pixel intensity along the full length of the connecting axis between the cells at the end of imaging. Visually apparent bands appeared in approximately 60% of the imaged cell pairs.

*Quantification window size.* To quantify the local ECM density we used a quantification window of the size of a cell diameter in all axes, in 2D simulations

(0.08 AU) or 3D in experiments (15 μm). This window size was set to optimize the tradeoff between including sufficient data versus too much irrelevant data within the window (Fig. S21). The number of pixels defining the quantification window (cell diameter in simulations, 15 μm in experiments) were calculated according to the transformed image resolutions. The same scale was also used upon shifting the quantification windows.

*Quantifying local fiber density in simulations.* The local fiber density was calculated as the accumulated fiber volume within the quantification window. We assume that the fiber volume is preserved even when the fiber is remodeled. However, this property does not hold in the simulated 2D representation of the fibers where their buckling property reduces the simulated fiber lengths. This is an inherent limitation of simulating a 3D process in 2D. To overcome this limitation we normalize each fiber to its initial length before summing the fiber length in the quantification window. More specifically, we considered two scenarios (Fig. S20E). (1) For the case where the fiber was located exclusively within the quantification window, its length at the onset of the simulation was used for quantification. (2) For the case where the fiber was not located exclusively within the quantification window (i.e., crossing the window boundaries), we used only the sub-fiber within the quantification window while adjusting to the full fiber length at the onset of the simulation: $Fiber_{Volume}(t) = Fiber_{InnerLength}(t) * \frac{Fiber_{Length}(t_0)}{Fiber_{Length}(t)}$, where $Fiber_{Volume}(t)$ is the fiber volume at time $t$, $Fiber_{InnerLength}(t)$ is the length of the sub-fiber within the quantification window at a time $t$, $Fiber_{Length}(t_0)$ is the overall fiber length at the onset of simulation and $Fiber_{Length}(t)$ is the overall fiber length at time $t$.

The fiber density within a quantification window was defined as the accumulated fiber volume within it. For single cells the mean fiber density of four windows, above, below, to the left and to the right of the cell was recorded (Fig. S20F).

*Quantifying local fiber density in experiments.* The mean fluorescent fibrin channel intensity was used as a proxy of fiber density within the transformed 3D quantification windows (see earlier). Quantification windows with over 5% of pixels extending beyond the image boundaries were marked as "invalid" and were excluded from further analysis. Quantification windows for single cells were calculated similarly to simulations, but in 3D, averaging the mean intensity in 32 orientations (see earlier).

*Normalizing the local fiber density in simulations and experiments.* To enable quantitative comparison across experiments and between simulations and experiments, we normalized the fiber density to its z-score - the number of standard deviations away from the mean background fiber density at quantification windows that were not influenced by the cells. Background quantification windows were defined for every location at the onset of simulation/imaging before (simulation) or where minimal (experiments) ECM remodeling occurred. To calculate the mean background fiber density we considered all quantification windows that did not intersect with the quantification window around the cells' center (Fig. S20G - simulations, windows step resolution = 0.02 cell diameter in each axis; Fig. S20H - experiments, windows step resolution = 1/10 of the image axis length for each axis). The mean (μ) and the standard deviation (σ) of all background quantification windows were calculated per simulation/location and were used to normalize each quantification window according to the z-score measure, $FiberDensity_{Z-score} = \frac{FiberDensity - \mu}{\sigma}$, i.e., the variation from the mean background intensity in units of standard deviation, where $FiberDensity$ is the fiber density quantification before normalization. This measure could be pooled and compared across locations, experiments and could even be used to compare simulations to experiments.

*Extracting local ECM density over time.* After performing the image transformation (see above), the quantification windows were placed adjacent to cell boundaries in 2D (simulations) or 3D (experiments) within the cuboid along the connecting axis between the cells, see Fig. S2 and Video S2. This location was updated at each time frame according to the current cell boundary positions that changed due to cell contraction (simulations) or motion (experiments). For example, this tracking corrected for axial drifting of cells toward the glass due to the contraction of the entire gel induced by cell forces. Shifts in the quantification windows were performed relative to this position. Offset in the Z axis translated to a quantification window placed above/below the pair-connected axis and perpendicular to the XY axis. Offset in the XY axis translated to a quantification window placed to left/right in XY without changing the Z position. Shifting the quantification window toward the communication partner or away from a single cell (window distance > 0) was performed on the connecting axis between the cells (Fig. 1C), or the axis defined by the image transformation angle in single cells. For a given time-lapse sequence we recorded and normalized the fiber density within the corresponding quantification windows over time.

We marked quantification windows within the time series as *invalid* if one of the following criteria holds: (1) Frames with annotated imaging artifacts within the corresponding imaging locations (bad cell segmentation, malfunctioning of the imaging microscope and "light waves", Video S7). (2) Overlapping quantification windows, in future corresponding time frames in the time series (Fig. S20I). (3)

Quantification windows with over 5% of pixels extending beyond the image boundaries (Fig. S20J).

*Eliminating imaging artifacts between time frames to avoid correlations not related to cell-ECM-cell communication.* Cells from the same experiment were embedded in the same fibrous gel inducing local ECM correlations that are not related to cell-ECM-cell communication. To eliminate these artifacts the local ECM density in each quantification window was normalized by two other windows (termed together "normalization border") located above and below (in the Z-axis) the quantification window (see Fig. 4C). Each normalization border height was 0.5 cell diameter and located with a gap of 0.25 cell diameter above or below the quantification window. These parameters were set to optimize the tradeoff between reducing erroneous correlations induced by local correlated ECM fluctuations near the quantification windows and maintaining the ECM fluctuations necessary for measuring cell-ECM-cell communication. Specifically, the parameters were determined to maximize same-versus-different analysis in live cells while minimizing it for dead cells. The normalized and corrected ECM density time series were used for all correlation-based analyses.

*Correlation-based analyses.* For correlation-based analysis we considered the longest sub-sequences with continuous valid time frames and mutual timestamps (Fig. S20K). We considered only cell pairs with mutual sub-sequences of at least 15 (temporal resolution = 15 mins) or 50 (temporal resolution = 5 mins) time frames. Correlation was calculated for the second derivative (simulations) or first derivative (experiments) of the fiber density dynamics according to stationarity criteria to avoid high correlations stemming from the monotonic increase of the fiber density (simulations and experiments) and its derivative (simulations) (Fig. S1, Fig. S5). We determined the first/second derivative detrending according to two stationarity tests Kwiatkowski–Phillips–Schmidt–Shin (KPSS)[54] and Augmented Dickey Fuller (ADF)[55]. The null hypothesis in the KPSS test is time series stationarity, while in the ADF test is time series non-stationarity. Temporal correlations were calculated using Pearson correlation on the derived time series.

*Same-versus-different pair analysis.* To establish that cell-ECM-cell communication of one cell pair can be distinguished from a second cell pair we tested whether the correlation between a cell pair ("same" pair) surpassed the correlation between one cell from that pair and another cell from a different cell pair ("different" pair) (Fig. 4A). This comparison was termed same-versus-different pair analysis. In this analysis, we considered all combinations of triplets of cells in an experiment, that included one communicating cell pair and another cell that takes part in another communicating pair (Fig. 4A). The quantification window of each cell in the analysis was always located in relation to its communication partner (Fig. 4A).

*Real-versus-fake pair analysis.* To control for misinterpreting non-communication related local ECM remodeling correlations as cell-ECM-cell communication we devised a standardized internal computational control that measures whether the correlation between a cell pair ("real" pair) surpassed the correlation between one cell from that pair and another "fake" cell (forming together a "fake" pair). The "fake" cell location was determined by following two constraints. The distance between the "fake" and its "real" partner is equal to the distance between the communicating cell pair, while maximizing the distance to other real cells. The first constraint was defined to allow comparable distance between the "real" and "fake" cells pairs. The second constraint was defined to minimize the effect of other cells in the vicinity. See depiction in Fig. 5A.

*Matchmaking analysis.* For each cell that takes part in a cell pair we tested our ability to identify its matched communication partner from all the other cells in that experiment (Fig. 5C). This was performed by ranking the ECM remodeling fluctuations (i.e., first derivative of the fiber density dynamics) correlations between the cell at test to all other potential communication partners and recording the ranking (i.e., the position in the sorted list of all correlations with other cells) of the true communication partner. The potential communication partners were determined based on the evaluation at test, for example considering only pairs with bands, pairs without bands, or all pairs regardless of having bands. To make this analysis independent of the number of cells in an experiment, we randomly selected, with repetition, 49 potential communication partners with an expected random probability of identifying the communication partner of 1/50 = 0.02. The purpose of the repetition in the random selection of the communication partners was to enable fair comparison between experiments that contained different numbers of communicating cells. The probability of identifying the true communication partner was recorded at the matchmaking score.

*Internal control to validate that our analysis is not an artifact of non-communication-related correlated local ECM remodeling fluctuations.* The contraction of the cells in the gel lead to local ECM correlations in the fibrous network, even in cell-free areas. To further verify that our results in the same-versus-different pair and matchmaking analyses were not merely an artifact of these local ECM correlations in the fibrous network, we compared the correlation of communicating cell pairs to ECM remodeling correlations in quantification windows that were placed in cell-free, fibrous areas. The intuition behind this control

experiment was to consider correlations in quantification windows that measure the proximity-component, or the local mutual ECM fluctuations, without the influence of the communicating cells. These controls were performed by manually annotating "fake" cell pairs, and analyzing them while "following" the corresponding "real" cells (Fig. S8A, cyan cells). Specifically, the quantification windows' locations followed their corresponding cell pair's motion by shifting in X′ and Y′ while maintaining a fixed distance from the real pair.

*Assessing sensitivity to temporal resolution.* To examine the sensitivity of our method to the temporal resolution we performed same-versus-different pair analysis for down-sampled time series. For the sake of completeness, all possible starting time frames were considered when setting the first time frame for sampling. For example, when down-sampling the temporal resolution from 5 to 15 mins per frame, there are 15/5 = 3 possible starting time frames: $t_0$, $t_1$ and $t_2$. When choosing $t_1$, for example, the first 3 sampled time frames are $t_1$, $t_4$, $t_7$. Time series shorter than 5 (temporal resolution = 15 mins) or 15 (temporal resolution = 5 mins) time frames were excluded from further correlation-based analysis. The initial chosen time frame for sampling was set for all same-versus-different computations of that time frame. For each temporal resolution, we pooled all same-versus-different results, which included multiple time series per down-sampled time series as described above. See Fig. S10B for full assessment.

*Pooling data across experiments for statistical assessment.* Each experiment was analyzed independently to avoid erroneous relations stemming from correlating two cells in different fibrous networks. Such correlations would be inherently lower when correlating ECM fluctuations between different networks versus in the same fibrous network, due to global network remodeling. Thus, correlating ECM fluctuations between different experiments will lead to erroneously optimistic results, which we avoided here by considering all possible combinations of "same"/"different", "real"/"fake", and matchmaking for each experiment independently. After analyzing each experiment independently we pooled all the results across experiments for statistical assessment. The non-parametric Wilcoxon signed-rank test was used for statistical analysis testing the null hypothesis that a distribution, such as of "same"-"different" correlations, was distributed around zero.

Summary of all measurements over all experimental conditions are reported in Table S1.

*Statistics and reproducibility.* All statistical data such as the test used, number of cells in the test, the distances between the cells, $p$-values obtained, etc. can be found in the relevant Fig. legend. Processed data is available and can be used for reproducibility.

*Leader-follower analysis with cross-correlation.* Finite-element simulations with a predefined leader and follower were examined using cross-correlation analysis, measuring the correlation between two time series under different time lags. We generated simulations where one cell in a pair was predetermined as the "leader" and its communication partner as the "follower". The "follower" cell "imitated" the "leader" cell contraction in the previous time step, thus the contraction of the "follower" cell is lagging one simulation round behind the "leader". The magnitude of influence that the leader had on the following was defined with the parameter $\alpha$: $Contraction_{follower}(t) = (1 - \alpha) * N(\mu, \sigma) + \alpha * Contraction_{leader}(t - 1)$, where $t > 1$, $\alpha = 0, 0.25, 0.5, 0.75$ or $1$, $\mu = 1$, $\sigma = 0.5$, and $Contraction_{leader}$ was drawn from the normal distribution $N(\mu, \sigma)$. The parameter $\alpha$ indicates the "followership" magnitude, higher values of $\alpha$ imply that the follower is more influenced by its leader contraction.

In a second simulation, we tested whether our approach can identify "leader" and "follower" even when the "follower" contracted more than the "leader", but was still copycating the leader's contraction with a time lag of 1 simulation round. The fold increase of the "follower's" contraction was defined using a parameter $\beta$: $Contraction_{follower}(t) = \beta * Contraction_{leader}(t - 1)$, where $\beta = 1, 1.05, 1.1$ or $1.2$.

Cross-correlation analysis was performed by comparing different time-lags to the cells' time series and evaluating in relation to the simulation ground truth. Cross-correlation was also performed on experimental data, where ground truth was not available, but did not identify any leader/follower relations, either due to lacking temporal resolution or lack of leaders/followers in our dataset.

*Data.* See Tables S2-S3 for detailed information regarding all simulated and experimental data in this work.

**Reporting summary.** Further information on research design is available in the Nature Portfolio Reporting Summary linked to this article.

## Data availability
Processed data and source code are publicly available at https://github.com/assafna/cell-ecm-project. The data include the processed ECM remodeling fluctuation time series for each cell, in the key simulations and experiments. The source code to perform all analyses presented here is included.

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

## Acknowledgements

This work was supported by the Israeli Council for Higher Education (CHE) via Data Science Research Center, Ben-Gurion University of the Negev, Israel (to A.Z.), by the Wellcome Leap Delta Tissue program (to A.Z.), the Israel Science Foundation (1474/16, to A.L.), the Israel Science Foundation- Israeli Centers for Research Excellence (1902/12, to A.L.), and the Zimin Institute for Engineering Solutions Advancing Better Lives (to A.L.). We thank Oren Tchaicheeyan, Nir Gov and Meghan Driscoll for critically reading the manuscript. We thank Jean-Yves Tinevez and Dmitry Ershov from Institut Pasteur for training AN in image acquisition and analysis, this training was made possible thanks to NEUBIAS (COST Action, CA15124) Short Term Scientific Mission (STSM).

## Author contributions

A.Z. and A.L. conceived the study. S.N. designed the experimental assay. S.N. and B.E. performed all experiments. Y.K. performed all simulations. S.G. designed the simulated fibrous networks. S.J. and M.E. provided the cancer cell line. A.N. developed analytic tools, analyzed, and interpreted the data. A.N., A.L., and A.Z. drafted the manuscript. G.M. and Y.T. assisted with analyses. A.K. assisted with the microscopy experiments. A.L. and A.Z. mentored the authors. All authors wrote and edited the manuscript and approved its content.

## Competing interests

The authors declare no competing interests.
