## [Peer Review File · Communications Biology]

Reviewers' comments:

Reviewer #1 (Remarks to the Author):

%%%%%%%% Summary

The clarifications given by the authors in the revision are very appreciated and have helped me understand key points in the manuscript. I am now convinced that the work and analyses that are presented are sufficient to make the authors' point: The method presented here can detect mechanical interactions between pairs of cells in fibrin gels by measuring temporal fluctuations of ECM density (termed cell-ECM-cell communication). It remains to be seen how well this method can be applied to other cell types and matrix compositions, but that can be addressed in future studies.

Revisions are needed to polish the manuscript text and make it more accessible to a general biological audience. Most of my points are suggestions to improve readability and make the manuscript accessible.

%%%%%%%% Major points

- 1) The ECM remodelling fluctuations are relatively small. Please add to the discussion how measurement errors (e.g. due to low spatial or temporal resolution) might affect the accuracy. The authors have already done experiments at different temporal resolutions, which they can use to explain this point.
- 2) Figure 3, D: Even among communicating cells there is a very large deviation. Does this mean that some cells are close enough to potentially communicate but behave as non-communicating cells? Or is there large timepoint-to-timepoint variation in otherwise clearly communicating cells?
- 3) Introduction page 3: The sentence starting with "The dictionary definition..." is disconnected from the flow of the rest of the text. I suggest to integrate it better with the sentence starting with "This form of long-range cell-cell force transmission..." by rephrasing.
- 4) Introduction page 3: "Such measurement lacks the sensitivity to measure the dynamic..." - It would be helpful to the reader if the reason why the measurements lack sensitivity would be briefly stated.
- 5) Introduction page 3: "Thus, current methods are limited in measuring potential cell-ECM-cell communication in the absence of visible bands...". A reader unfamiliar with the concept of band formation will be left confused. I suggest explaining the temporal order of events that are observed in experiments (while citing appropriate literature). First, cells contract and thereby densify nearby ECM. Then, sometimes between pairs of cells a 'bridge' of densified ECM forms over a long range connecting the two cells. The authors could use this summary to strengthen the argument for developing more sensitive measurement methods that do not rely on band formation, because arguably band formation is only the final element in the temporal sequence of events. It would also be helpful to indicate timescales (cell-ECM-cell communication happens in the timescale of minutes, band formation in the timescale of hours?).
- 5) Results page 4: "While these simulations do not reflect the true complexity of biological systems, they capture the essence of the mechanical elements of cell contractility and force propagation in fibrous nonlinear elastic networks [...]." This sentence needs to be underpinned with citations to previous work or general reviews (e.g. Broedersz and MacKintosh. Rev Mod Phys 86.3 (2014): 995.)
- 6) Results page 4: "To enable quantitative comparison between simulations, we normalized the fiber density to its z-score [...]." It would be helpful to have a simple example for the unfamiliar reader. For example: This means that if the fibers are denser than average, the z-score will be > 0 , while for fibers less dense than average, the z-score < 0 .
- 7) Figure 1, panel C: The legend explains that window distance is bounded, but it does not clarify what

the window distance itself is. Consider using the cartoon in Supplementary figure S3 panel A in Figure 1 as well.

8) Results page 7: "We expected the temporal correlation of ECM remodeling fluctuations [...]" I understand this concept now after reading the manuscript several times, but an unfamiliar reader will be hopelessly lost. I suggest to explain in simple terms what this means, e.g.: Cell-ECM pulling happens in short noisy bursts of activity over time. This random remodeling noise leads to fluctuations in ECM density over time. We wondered if ECM remodeling fluctuations would correlate in communicating cells (i.e. communicating cells pull on ECM in near-synchrony, possibly reacting to each other's pull).

9) Figure 4 B (and other similar plots). It would be extremely helpful to show a cartoon example of how such a plot would look like for only two pairs of cells (cell A, cell B, cell C, cell D, where A=B are a pair and C=D are a pair). Understanding how to interpret these plots is crucial to the paper. I have attached a sketch of what I mean.

10) Results page 16: "[...] analysis that considered ECM regions (without cells) located close to each other ("fake pairs [...])". A sentence similar to this would be helpful to mention on page 15, when the concept of a fake cell is first introduced. On that note, this could be indicated directly in figure 5 A, by writing "fake cell = cell-free ECM region".

11) Results page 22: "cell-ECM-cell communication exists even when no visible band is formed". Maybe I missed it, but what is the proportion of communicating pairs out of all pairs within pulling distance that you observed?

12) Discussion page 28: "[...] captures the essence of mechanical elements of cell contractility [...]" Please add citations to previous modelling work to underpin this statement.

13) Discussion page 29: "Until now, the consensus in the field was [...]" Provide citations.

14) Discussion page 29: "Our method [...] correlates local ECM deformations [...]" The word "deformations" is imprecise, as it implies the ECM is deformed, but it could also simply be displaced without undergoing deformation. Please reword.

15) Discussion page 29: "[...] can robustly identify the leader and follower cells [...]" Please specify that this statement only applies to simulations.

16) Methods page 29: "the average connectivity of the network was set to eight [...]" This is a relatively high number (collagen gels have an average connectivity around 3-4, see e.g. Burla et al. PNAS 117.15 (2020): 8326-8334.). I expect the simulated network to be stiffer as a result. Please briefly discuss.

Minor points

1) Please add line numbers when sending the manuscript for review.

2) Introduction, page 3 first sentence: The authors cite references that ECM deformations can propagate tens of cell diameters away. What distance of propagation have they observed in their own simulations? What about the experiments? This could be briefly commented on in the discussion.

3) Introduction page 3: The dictionaries are cited but are not listed in the reference list.

4) Introduction page 3: "Most current measurements characterize the structure of the fibrous band [...]" - does this sentence refer to the citations given in the previous sentence?

- 5) Figure 1, C: Mention at what simulated time the measurement was made (at 50% contraction?)
- 6) Figure 2, E and F: individual data points are shown overlaid to the boxplots for communicating cells, but not for non-communicating cells. This also happens elsewhere in the manuscript. Please make the plots consistent.
- 7) Figure 2 legend: typo in the sentence "The variability in fiber density in is a consequence [...]" - there is an extra "in".
- 8) Figure 2 legend: typo in the sentence "Each data point records the correlation between a simulated communicating cell pair, each box plots record the [...]", "box plots" should be singular "box plot"
- 9) Results pages 10-11: typo in "These results conclude that the formation of dense fibrous bands between fibroblasts cell pairs [...]", "fibroblasts" should be singular "fibroblast"
- 10) Figure 3, C: Mention the time of the measurement.
- 11) Results page 12 typo: "[...] whether the ECM fluctuations of one cells pair has a [...]" the word 'cells' should be singular 'cell'
- 12) Results page 1: "We next aimed at extending our simulation results [...]" The computational analysis was extended, not the simulation results.
- 13) Results page 19: "As negative controls we included [...]". Negative controls for what? Please reword to clarify. Maybe you could start with the sentence that appears later in the text "[...] a validation that our measurement for cell-ECM-cell communication is robust [...]".
- 14) Figure 6 legend: typo in line 3, "where" instead of "were"
- 15) Methods page 32: typo "Finite element simulations of cells contraction in fibrous networks". The word "cells" should be singular "cell".
- 16) Methods page 35: Exponents are not shown as superscript (e.g. "8x103", the final 3 should be an exponent indicating thousands of cells).
- 17) Methods page 36: Atomic numbers in molecules are not written as subscripts, e.g. for carbon dioxide (CO₂).
- 18) Methods page 38: The authors probably intended "defective" instead of "defected".

Assaf Zaritsky, Ph.D.
Assistant Professor

June 29th, 2023

We would like to thank the reviewer who thoroughly read our (long) manuscript and provided, for the second time, constructive feedback that helped us improve the manuscript.

Below please find our point-by-point response to the Reviewer's latest comments.

Reviewer's Comments

Review of "Inference of long-range cell-cell mechanical communication from ECM remodeling fluctuations" by Nahum et al.

Summary

The clarifications given by the authors in the revision are very appreciated and have helped me understand key points in the manuscript. I am now convinced that the work and analyses that are presented are sufficient to make the authors' point: The method presented here can detect mechanical interactions between pairs of cells in fibrin gels by measuring temporal fluctuations of ECM density (termed cell-ECM-cell communication). It remains to be seen how well this method can be applied to other cell types and matrix compositions, but that can be addressed in future studies.

Revisions are needed to polish the manuscript text and make it more accessible to a general biological audience. Most of my points are suggestions to improve readability and make the manuscript accessible.

We thank the reviewer for this detailed and constructive feedback.

Major comments

1. The ECM remodelling fluctuations are relatively small. Please add to the discussion how measurement errors (e.g. due to low spatial or temporal resolution) might affect the accuracy. The authors have already done experiments at different temporal resolutions, which they can use to explain this point.

We included in the Discussion the following text: “*The ECM remodeling fluctuations are relatively small. Noise and measurement errors, for example due to low spatial or temporal resolution, might deteriorate the communication signal and hamper our ability to measure cell-ECM-cell communication. While this is a possibility, validations in our experimental settings, by artificially reducing temporal (Fig. S10) or spatial (Fig. S21) resolution demonstrated that the communication signal is quite robust.*”

2. Figure 3, D: Even among communicating cells there is a very large deviation. Does this mean that some cells are close enough to potentially communicate but behave as non-communicating cells? Or is there large timepoint-to-timepoint variation in otherwise clearly communicating cells?

Formation of densified bands between cell pairs is a highly heterogeneous process. Bands between different cell pairs initiate at different timings and reach different densities. However, on average, the differences in fiber density between cell pairs and around single cells is significant. This heterogeneity is the main reason for the large standard deviation seen in fig. 3D.

Importantly, our method’s ability to identify cell-ECM-cell communication, via correlation in ECM remodeling dynamics between cell pairs, is decoupled from band formation as demonstrated, for example, in cell pairs without visible bands or by identifying the sweet spot for the quantification windows in the less dense regions above the connecting axis. Figure S14 extensively assesses the interplay between fiber density, change in fiber density in time and the ability to measure cell-ECM-cell communication.

3. Introduction page 3: The sentence starting with “The dictionary definition...” is disconnected from the flow of the rest of the text. I suggest to integrate it better with the sentence starting with “This form of long-range cell-cell force transmission...” by rephrasing.

We integrated this introductory part according to the reviewer’s suggestion “*This form of long-range cell-cell force transmission through the ECM can be viewed as the imparting or exchanging of information between cells, and thus is aligned with the definition of communication {Stevenson, 2010 #1553} termed here cell-ECM-cell communication. This mode of long-range mechanical cell-ECM-cell communication was shown to coordinate various biological processes...*”.

4. Introduction page 3: “Such measurement lacks the sensitivity to measure the dynamic...” - It would be helpful to the reader if the reason why the measurements lack sensitivity would be briefly stated.

To make this point more clear we integrated the aforementioned sentence with the following one “*Such measurement lacks the sensitivity to measure the dynamic reciprocal mechanical information transfer between the cells by excluding potential cell-ECM-cell communication in*

the absence of visible bands, thus hampering our ability to distinguish which cells are actually communicating from the many cells that have the potential to communicate.”

5. Introduction page 3: “Thus, current methods are limited in measuring potential cell-ECM-cell communication in the absence of visible bands...”. A reader unfamiliar with the concept of band formation will be left confused. I suggest explaining the temporal order of events that are observed in experiments (while citing appropriate literature). First, cells contract and thereby densify nearby ECM. Then, sometimes between pairs of cells a ‘bridge’ of densified ECM forms over a long range connecting the two cells. The authors could use this summary to strengthen the argument for developing more sensitive measurement methods that do not rely on band formation, because arguably band formation is only the final element in the temporal sequence of events. It would also be helpful to indicate timescales (cell-ECM-cell communication happens in the timescale of minutes, band formation in the timescale of hours?).

We start the Introduction by describing the band formation process, before describing quantification methods. We now included the time-scales, while citing appropriate literature, as the reviewer suggested: *“When embedded in fibrous biological hydrogels, such as collagen or fibrin, cells contract and thereby deform and densify nearby ECM fibers. Then, in the time-scale of a few hours, these remodeling can form a visible fibrous band of aligned and dense fibers coupling neighboring cells mechanically which can influence the cells’ internal molecular state {Doyle, 2016 #1408} and active response {Pakshir, 2019 #1409}.”*

6. Results page 4: “While these simulations do not reflect the true complexity of biological systems, they capture the essence of the mechanical elements of cell contractility and force propagation in fibrous nonlinear elastic networks [...]” This sentence needs to be underpinned with citations to previous work or general reviews (e.g. Broedersz and MacKintosh. Rev Mod Phys 86.3 (2014): 995.)

We now include appropriate citations. Moreover, since we are not implicitly modeling the cytoskeleton or stress fibers inside the cell, we revised the sentence to: *“...captures the essence of cell contraction and force propagation in fibrous nonlinear elastic networks {Schwarz, 2013 #1555;Notbohm, 2015 #1208;Liang, 2016 #1476;Ronceray, 2016 #1404;Sopher, 2018 #990;Goren, 2020 #1403} and thus...”*

7. Results page 4: “To enable quantitative comparison between simulations, we normalized the fiber density to its z-score [...]”. It would be helpful to have a simple example for the unfamiliar reader. For example: This means that if the fibers are denser than average, the z-score will be > 0 , while for fibers less dense than average, the z-score < 0 .

We revised the text accordingly: *“To enable quantitative comparison between simulations, we normalized the fiber density to its z-score - the number of standard deviations away from the mean background fiber density at regions that were not influenced by the cells, implying positive*

z-scores values where the fibers are denser than average and negative values for regions that are less dense than the average background fiber density (Methods).“

8. Figure 1, panel C: The legend explains that window distance is bounded, but it does not clarify what the window distance itself is. Consider using the cartoon in Supplementary figure S3 panel A in Figure 1 as well.

We moved the cartoon from Fig. S3A to Fig. 1C.

Revised Fig. 1:

9. Results page 7: “We expected that the temporal correlation of ECM remodeling fluctuations [...]” I understand this concept now after reading the manuscript several times, but an unfamiliar reader will be hopelessly lost. I suggest to explain in simple terms what this means, e.g.: Cell-ECM pulling happens in short noisy bursts of activity over time. This random remodeling noise leads to fluctuations in ECM density over time. We wondered if ECM remodeling fluctuations would correlate in communicating cells (i.e. communicating cells pull on ECM in near-synchrony, possibly reacting to each other’s pull).

We revised the text accordingly “*Cell-ECM pulling occurs in noisy bursts of activity leading to fluctuations in ECM density that propagate to large distances. We hypothesized that these ECM remodeling fluctuations, measured in quantification windows adjacent to each cell, would temporally correlate between communicating cell pairs due to synchronization response of the mechanically coupled matrix, exceeding the temporal correlation between non-communicating cell pairs (Fig. 2A).*”

10. Figure 4 B (and other similar plots). It would be extremely helpful to show a cartoon example of how such a plot would look like for only two pairs of cells (cell A, cell B, cell C, cell D, where A=B are a pair and C=D are a pair). Understanding how to interpret these plots is crucial to the paper. I have attached a sketch of what I mean.

Revised Fig. 4:

11. Results page 16: “[...] analysis that considered ECM regions (without cells) located close to each other (“fake pairs [...])”. A sentence similar to this would be helpful to mention on page 15, when the concept of a fake cell is first introduced.

We revised the text accordingly “*To control for potential masking of cell-ECM-cell communication by correlated non-communication related local ECM remodeling we devised a computational control that considered ECM regions (without cells) located close to a communicating cell, that we term real-versus-fake pair analysis. We created new pairs where*

each is composed of one real cell and another fake cell (cell-free ECM region) located in the exact same distance as the matched pair of communicating cells, and away from other cells (Fig. 5A, Methods).”

On that note, this could be indicated directly in figure 5 A, by writing “fake cell = cell-free ECM region”.

We indicated “fake cell = cell-free ECM region” in Fig. 5A.

Revised Fig. 5:

12. Results page 22: “cell-ECM-cell communication exists even when no visible band is formed”. Maybe I missed it, but what is the proportion of communicating pairs out of all pairs within pulling distance that you observed?

This is shown in Fig.7 H-I. 80%/95% for standard/rapid imaging correspondingly. We adjusted the title of panels H and I (“standard/fast imaging (no band)”) to emphasize that they refer to the full figure panel.

Revised Fig. 7:

13. Discussion page 28: “[...] captures the essence of the mechanical elements of cell contractility [...]” Please add citations to previous modelling work to underpin this statement.

We included citations as requested.

14. Discussion page 29: “Until now, the consensus in the field was [...]” Provide citations.

We included citations as requested.

15. Discussion page 29: “Our method [...] correlates local ECM deformations [...]” The word “deformations” is imprecise, as it implies the ECM is deformed, but it could also simply be displaced without undergoing deformation. Please reword.

We replaced the term “deformations” with “remodeling” throughout the manuscript.

16. Discussion page 29: “[...] can robustly identify the leader and follower cells [...]” Please specify that this statement only applies to simulations.

The sentence starts with the statement that these experiments are simulations, nevertheless, we included the word “simulated” another time to make sure this is not missed by the reader: “*We simulated a situation where one cell influenced its communicating partner, determining its future contractions and demonstrated that our method can robustly identify the leader and follower cells from the simulated ECM-remodeling fluctuations.*”

17. Methods page 29: “the average connectivity of the network was set to eight [...]”. This is a relatively high number (collagen gels have an average connectivity around 3-4, see e.g. Burla et al. PNAS 117.15 (2020): 8326-8334.). I expect the simulated network to be stiffer as a result. Please briefly discuss.

The reviewer is correct that the connectivity value of our simulated networks is relatively high, above the characteristic value of collagen networks. In our previous numerical studies using similar networks, we evaluated the effect of network connectivity on force transmission by contractile cells, and reached the conclusion that the main characteristics of force transmission are maintained in networks of different connectivities. Therefore, we believe that the trends presented in the manuscript should not be strongly affected by the exact value of network connectivity. We choose a higher level of connectivity to simplify the finite element simulations and accommodate numerical convergence issues. In the revised manuscript we added the following text to the Method section to explain our choice of connectivity: “*We note, however, that the mean connectivity of collagen networks is typically lower and within the range of 3-4 {Burla, 2020 #1556}. In our previous numerical studies, we found that the main characteristics of force transmission by contractile cells are maintained in the low and higher connectives networks {Sopher, 2018 #990;Goren, 2020 #1403}.*”

Minor comments

1. Please add line numbers when sending the manuscript for review.

Done.

2. Introduction, page 3 first sentence: The authors cite references that ECM deformations can propagate tens of cell diameters away. What distance of propagation have they observed in their own simulations? What about the experiments? This could be briefly commented on in the discussion.

We agree with the reviewer that the range of cell-cell interaction is an important aspect to discuss. In our simulations, we detected a communication signal in cell pairs when cells were located up to 9 cell-diameters apart (Fig. 4B). In experiments of fibroblasts in fibrin gels, we detected a similar distance of 8-10 cell-diameters on which a communication signal was detected (Figure 4D). We note that we have not simulated or analyzed larger cell-cell distances in our study, and thus these ranges are lower bounds. Accordingly, we included the following text to the revised Discussion: *“The method quantifies the local fiber remodeling dynamics between pairs of communicating cells, located up to a distance of 10 cell-diameters apart, in 2D (simulations) and 3D (experiments) by applying the following key steps.”*

3. Introduction page 3: The dictionaries are cited but are not listed in the reference list.

Done.

4. Introduction page 3: “Most current measurements characterize the structure of the fibrous band [...]” - does this sentence refer to the citations given in the previous sentence?

We have updated the most relevant citations to this sentence, the ones that focus on analyzing the structure of the fibrous band (i.e., density and alignment of fibers): *“Most current measurements characterize the structure of the fibrous band extending between mechanically coupled cells to inform on cell-cell mechanical coupling, while relying on the visibility of a band between cells, formed when cell-generated forces are strong enough {Shi, 2014 #969; Kim, 2017 #1076; Ban, 2018 #1402; Natan, 2020 #1207; Doha, 2022 #1550}”*.

5. Figure 1, C: Mention at what simulated time the measurement was made (at 50% contraction?)

Done. Figure legend now reads *“Quantifying fiber densification after 50% cell contraction as a function of the distance around”*

6. Figure 2, E and F: individual data points are shown overlaid to the boxplots for communicating cells, but not for non-communicating cells. This also happens elsewhere in the manuscript. Please make the plots consistent.

We chose not to display all data points for non-communicating cell pairs because there are too many of them, while the number of data points for communicating cells is much smaller and thus can be plotted. We included a sentence to explain this point in the corresponding figure legends. E.g., *“Data points for non-communicating cell pairs were not displayed in panels E-F because there are too many of them, while the number of data points for communicating cells is much smaller and thus can be plotted.”*

7. Figure 2 legend: typo in the sentence “The variability in fiber density in is a consequence [...]” - there is an extra “in”.

Corrected.

8. Figure 2 legend: typo in the sentence “Each data point records the correlation between a simulated communicating cell pair, each box plots record the [...]”, “box plots” should be singular “box plot”.

Corrected.

9. Results pages 10-11: typo in “These results conclude that the formation of dense fibrous bands between fibroblasts cell pairs [...]”, “fibroblasts” should be singular “fibroblast”.

Corrected.

10. Figure 3, C: Mention the time of the measurement.

Done. Figure legend now reads: “*Quantifying cell-ECM-cell communication after (slightly over) four hours of live cell imaging as a function of distance between...*”.

11. Results page 12 typo: “[...] whether the ECM fluctuations of one cells pair has a [...]” the word ‘cells’ should be singular ‘cell’.

Corrected.

12. Results page 1: “We next aimed at extending our simulation results [...]” The computational analysis was extended, not the simulation results.

Revised to “We next aimed at extending our computational analysis of simulated data to distinguishing between pairs of communicating cells in experimental data”

13. Results page 19: “As negative controls we included [...]”. Negative controls for what? Please reword to clarify. Maybe you could start with the sentence that appears later in the text “[...] a validation that our measurement for cell-ECM-cell communication is robust [...]”.

We revised the text accordingly: “*To verify that ECM-remodeling correlations are robust to ECM fluctuations induced by other cells in the gel we included a new set of experiments as negative controls.*”

14. Figure 6 legend: typo in line 3, “where” instead of “were”.

Corrected.

15. Methods page 32: typo “Finite element simulations of cells contraction in fibrous networks”. The word “cells” should be singular “cell”.

Corrected.

16. Methods page 35: Exponents are not shown as superscript (e.g. “8x103”, the final 3 should be an exponent indicating thousands of cells).

Corrected.

17. Methods page 36: Atomic numbers in molecules are not written as subscripts, e.g. for carbon dioxide (CO₂).

Corrected.

18. Methods page 38: The authors probably intended “defective” instead of “defected”.

Corrected.

We look forward to your evaluation of this submission.

Best regards,

Assaf Zaritsky, Ph.D.